

Local time extent of magnetopause reconnection X-lines using space-ground coordination
Ying Zou12, Brian M. Walsh3, Yukitoshi Nishimura45, Vassilis Angelopoulos6, J.
Michael Ruohoniemi7, Kathryn A. McWilliams8, Nozomu Nishitani9
1. Department of Astronomy and Center for Space Physics, Boston University, Massachusetts,
USA
2. Cooperative Programs for the Advancement of Earth System Science, University Corporation
for Atmospheric Research, Boulder, Colorado, USA
3. Department of Mechanical Engineering and Center for Space Physics, Boston University,
Boston, Massachusetts, USA
4. Department of Electrical and Computer Engineering and Center for Space Sciences, Boston
University, Boston, Massachusetts, USA
5. Department of Atmospheric and Oceanic Sciences, University of California, Los Angeles,
California, USA
6. Department of Earth, Planetary and Space Sciences, University of California, Los Angeles,
California, USA
7. The Bradley Department of Electrical and Computer Engineering, Virginia Tech, Blacksburg,
Virginia, USA
8. Institute of Space and Atmospheric Studies, University of Saskatchewan, Saskatoon,
Saskatchewan, Canada
9. Center for International Collaborative Research, Institute for Space-Earth Environmental
Research, Nagoya University, Nagoya, Japan

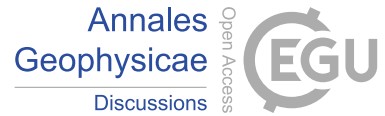

Corresponding author: Ying Zou
1. Department of Astronomy and Center for Space Physics, Boston University, Massachusetts,
USA
2. Cooperative Programs for the Advancement of Earth System Science, University Corporation
for Atmospheric Research, Boulder, Colorado, USA
yingzou@bu.edu
Keyword: 2784 Solar wind–magnetosphere interactions; 2724 Magnetopause, cusp, and
boundary layers; 7835 Magnetic reconnection



Abstract
Magnetic reconnection X-lines can vary considerably in length. At the Earth's magnetopause,
the length generally corresponds to the extent in local time. The extent has been probed by multi-
spacecraft crossing the magnetopause, but the estimates have large uncertainties because of the
assumption of a continuous X-line between spacecraft and the lack of information beyond areas of
spacecraft coverage. The extent has also been inferred by radars as fast ionospheric flows moving
anti-sunward across the open-closed field line boundary, but whether a particular ionospheric flow
results from reconnection needs to be confirmed. To achieve a reliable interpretation, we compare
X-line extents probed by multi-spacecraft and radars for three conjunction events. We find that
when reconnection is active at only one spacecraft, only the ionosphere conjugate to this spacecraft
shows a channel of fast anti-sunward flow. When reconnection is active at two spacecraft and the
spacecraft are separated by <1 Re, the ionosphere conjugate to both spacecraft shows a channel of
fast anti-sunward flow. The consistency allows us to determine the X-line extent by measuring the
ionospheric flows. The flow extent is 520, 572, and 1260 km, corresponding to an X-line extent
of 4, 5, and 11 Re. This strongly indicates that both spatially patchy (a few Re) and spatially
continuous and extended reconnection (>10 Re) are possible forms of reconnection at the
magnetopause. Interestingly, the extended reconnection develops from a localized patch via
spreading across local time. Potential effects of IMF Bx and By on the X-line extent are discussed.







## 1. Introduction

A long-standing question in magnetic reconnection is what is the spatial extent of reconnection in the direction normal to the reconnection plane. At the Earth's magnetopause, for a purely southward IMF, this corresponds to the extent in the local time or azimuthal direction. The extent of reconnection has significant relevance to solar wind-magnetosphere coupling, as it controls the amount of energy being passed through the boundary from the solar wind into the magnetosphere and ionosphere. Magnetopause reconnection tends to occur at sites of strictly anti-parallel magnetic fields as anti-parallel reconnection [e.g. *Crooker,* 1979; *Luhmann et al.*, 1984], or occur along a line passing through the subsolar region as component reconnection [e.g. *Sonnerup*, 1974; *Gonzalez and Mozer*, 1974]. This, however, does not represent the extent of active reconnection X-lines, as reconnection may not be active at all portions of this configuration, but can occur at discontinuous patches or over a limited segment only.

Numerical models show that reconnection tends to occur at magnetic separators, i.e. at the junction between regions of different magnetic field topologies, and global MHD models have identified a spatially continuous separator along the magnetopause [*Dorelli et al.*, 2007; *Laitinen et al.*, 2006, 2007; *Haynes and Parnell*, 2010; *Komar et al.*, 2013; *Glocer et al.*, 2016]. However, little is known about where and over what range along the separators reconnection is active. Reconnection in numerical simulations can be activated by introducing perturbations of the magnetic field or can grow spontaneously with instability or resistivity inherent in the system [e.g. *Hesse et al.*, 2001; *Scholer et al.*, 2003]. When reconnection develops as patches (as due to the instabilities or localized perturbations), the patches can spread in the direction out of the reconnection plane [*Huba and Rudakov,* 2002; *Shay et al.* 2003; *Lapenta et al.*, 2006; *Nakamura et al.*, 2012; *Shepherd and Cassak*, 2012; *Jain et al.*, 2013]. The patches either remain patchy after



spreading if the current layer is thick, or form an extended X-line if the current layer is already
thin [*Shay et al.*, 2003].
The extent of reconnection X-lines has been observationally determined based on fortuitous
satellite conjunctions where the satellites detect signatures of active reconnection at the
magnetopause at different local times nearly simultaneously [*Phan et al.*, 2000, 2006; *Walsh et*
*al.*, 2014a, 2014b, 2017]. The satellites were separated by a few Re in *Phan et al.* [2000] and *Walsh*
*et al.* [2014a, 2014b, 2017], and >10 Re in *Phan et al.* [2006], and this is interpreted as the X-line
being longer than a few Re and even 10 Re, respectively. At the magnetopause, X-lines of a few
Re are often referred to as spatially patchy [e.g., *Fear et al.*, 2008, 2010], and X-lines of >10 Re
are spatially extended [*Dunlop et al.*, 2011; *Hasegawa et al.*, 2016]. The extent of X-lines has been
alternatively determined by studying the structures of newly reconnected flux tubes, i.e., flux
transfer events (FTEs) [*Russell and Elphic*, 1978; *Haerendel et al.*, 1978]. Conceptual models
regard FTEs either as azimuthally narrow flux tubes that intersect the magnetopause through nearly
circular holes, as formed by spatially patchy X-lines [*Russell and Elphic*, 1978], or as azimuthally
elongated bulge structures or flux ropes that extend along the magnetopause, as formed by spatially
extended X-lines [*Scholer,* 1988; *Southwood et al.,* 1987; *Lee and Fu,* 1985]. FTEs have been
observed to be > or <2 Re wide in local time [*Fear et al.*, 2008, 2010; *Wang et al.*, 2005, 2007].
FTEs have even been observed across ~20 Re from the subsolar region to the flanks [*Dunlop et*
*al.*, 2011]. But it is unclear whether these FTEs are branches of one extended bulge or flux rope,
or multiple narrow tubes formed simultaneously. When the satellites are widely spaced, it is in
general questionable whether an X-line/FTE is spatially continuous between the satellites or
whether satellites detect the same moving X-line/FTE. Satellites with a small separation may
possibly measure the same X-line/FTE, but only provide a lower limit estimate of the extent. An



X-line/FTE may also propagate or spread between satellite detection but satellite measurements
cannot differentiate the spatial and temporal effects.
This situation can be improved by studying ionospheric signatures of reconnection and FTEs,
since their spatial sizes in the ionosphere can be obtained from wide field ground instruments or
Low-Earth orbit spacecraft. The ionospheric signatures include poleward moving auroral forms
(PMAFs), channels of fast flows moving anti-sunward across the open-closed field line boundary
[e.g., *Southwood*, 1985], and cusp precipitation [*Lockwood and Smith*, 1989, 1994; *Smith et al.*,
1992]. Radar studies have shown that the flows can differ considerably in size, varying from tens of
km [*Oksavik et al.*, 2004, 2005], to hundreds of km [*Goertz et al.*, 1985; *Pinnock et al.*, 1993, 1995;
*Provan and Yeoman*, 1999; *Thorolfsson et al.*, 2000; *McWilliams et al.*, 2001a, 2001b], and to
thousands of km [*Provan et al.*, 1998; *Nishitani et al.*, 1999; *Provan and Yeoman*, 1999]. A
similarly broad distribution has been found for PMAFs [e.g. *Sandholt et al.*, 1986, 1990; *Lockwood*
*et al.*, 1989, 1990; *Milan et al.*, 2000, 2016] and the cusp [*Crooker et al.*, 1991; *Newell and Meng*,
1994; *Newell et al.*, 2007]. This range of spatial sizes in the ionosphere approximately corresponds
to a range from <1 to >10 Re at the magnetopause. However, care needs to be taken when
interpreting the above ionospheric features, since they could also form due to other drivings such
as solar wind dynamic pressure pulses [*Lui and Sibeck*, 1991; *Sandholt et al.*, 1994]. An
unambiguous proof of their connection to magnetopause reconnection requires simultaneous
space-ground coordination [*Elphic et al.,* 1990; *Denig et al.,* 1993; *Neudegg et al.,* 1999, 2000;
*Lockwood et al.*. 2001; *Wild et al.,* 2001, 2005, 2007; *McWilliams et al.*, 2004; *Zhang et al.,* 2008].
Therefore a reliable interpretation of reconnection X-line extent has been difficult due to
observation limitations. We will address this by comparing X-line extents probed by multi-
spacecraft and radars using space-ground coordination. On one hand, this enables us to investigate



whether X-lines are continuous between satellites, and how wide X-lines extends beyond satellites.
On the other hand, this helps to determine whether reconnection is the driver of ionospheric
disturbances and whether the in-situ extent is consistent with the ionospheric disturbance extent.
It may be noteworthy to point out that we only address the X-line extent in the local time
direction, similarly to previous observations. If the X-line has a tilted orientation relative to the
equatorial plane, the local time extent will be shorter than the total extent.  How X-lines tilt is a
subject of ongoing research. Various models have been proposed to predict the tilt [*Alexeev et*
*al.*, 1998; *Moore et al.*, 2002; *Trattner et al.*, 2007; *Swisdak and Drake*, 2007; *Borovsky*, 2013;
*Hesse et al.*, 2013] but their performance is still under test [e.g., *Komar et al*., 2015]. The local
time extent is what determines the amount of magnetic flux opened in the solar wind-
magnetopshere coupling [e.g. *Newell et al.*, 2007].

2. Methodology
We use conjugate measurements between the Time History of Events and Macroscale
Interactions during Substorms (THEMIS) [*Angelopoulos*, 2008] and Super Dual Auroral Network
(SuperDARN) [*Greenwald et al.*, 1995]. We focus on intervals when the IMF in OMNI data
remains steadily southward. We require that two of the THEMIS satellites fully cross the
magnetopause nearly simultaneously and that the satellite data provide clear evidence for
reconnection occurring or not. The full crossings are identified by a reversal of the Bz magnetic
field and a change in the ion energy spectra. The requirements of nearly simultaneous crossings
and steady IMF conditions help to reduce the spatial-temporal ambiguity by satellite measurements,
where the presence/absence of reconnection signatures at different local times likely reflects
spatial structures of reconnection. Reconnection can still possibly vary between the two satellite
crossings, and we use the radar measurements to examine whether the reconnection X-line of





interest has continued to exist and maintained its spatial size.
Fluid (MHD) evidence of magnetopause reconnection includes plasma bulk flow acceleration
at the magnetopause. This acceleration should be consistent with the prediction of tangential stress
balance across a rotational discontinuity, i.e. Walen relation [*Hudson*, 1970; *Paschmann et al.*,
1979]. The Walen relation is expressed as
$$\Delta V_{predicted} = \pm(1 - \alpha_1)^{1/2}(\mu_o \rho_1)^{-1/2}[B_2(1 - \alpha_2)/(1 - \alpha_1) - B_1] \qquad (1)$$
Where $\Delta V$ is the change in the plasma bulk velocity vector across the discontinuity. $B$ and $\rho$ are
the magnetic field vector and plasma mass density. $\mu_0$ is the vacuum permeability. $\alpha = (p_\parallel -$
$p_\perp)\mu_0/B^2$ is the anisotropy factor where $p_\parallel$ and $p_\perp$ are the plasma pressures parallel and
perpendicular to the magnetic field. The magnetic field and plasma moments are obtained from
the fluxgate magnetometer (FGM) [*Auster et al.*, 2008] and the ElectroStatic Analyzers (ESA)
instrument [*McFadden et al.*, 2008]. The plasma mass density is determined using the ion number
density, assuming a mixture of 95% protons and 5% helium. The subscripts 1 and 2 refer to the
reference interval in the magnetosheath and to a point within the magnetopause, respectively. The
magnetosheath reference interval is a 10-s time period just outside the magnetopause. The point
within the magnetopause is taken at the maximum ion velocity change across the magnetopause.
We ensure that the plasma density at this point is >20% of the magnetosheath density to avoid the
slow-mode expansion fan [*Phan et al.*, 1996]. We compare the observed ion velocity change with
the prediction from the Walen relation. The level of agreement is measured by $\Delta V^* =$
$\Delta V_{obs} \cdot \Delta V_{predicted} \Big/ |\Delta V_{predicted}|^2$ , following *Paschmann et al.* [1986]. Here $\Delta V_{obs}$ is the
observed ion velocity change.
A kinetic signature of reconnection is found as D-shaped ion distributions at the magnetopause.
As magnetosheath ions encounter newly opened magnetic field lines at the magnetopause, they



either transmit through the magnetopause entering the magnetosphere or reflect at the boundary.
The transmitted ions have a cutoff parallel velocity (i.e. de-Hoffman Teller velocity) below which
no ions could enter the magnetosphere. The D-shaped ion distributions persist from the active
reconnection region at the magnetopause into the ionosphere where they appear as cusp ion steps
[*McWilliams et al.*, 2004]. We require the satellites to operate in the Fast Survey or Burst mode in
which ion distributions are available at 3 s resolution.

We determine reconnection being active if the plasma velocity change across the magnetopause

is consistent with the Walen relation with $\Delta V^* >= 0.5$, and if the ions at the magnetopause show a
D shape distribution. Reconnection is deemed absent if neither of the two signatures is detected.
We require that at least one of the two satellites observe reconnection signatures. Reconnection is
regarded as ambiguous if only one of the two signatures is detected, and such reconnection is
excluded from our analysis.

We mainly use the three SuperDARN radars located at Rankin Inlet (RKN, geomagnetic 72.6°

MLAT, -26.4° MLON), Inuvik (INV, 71.5° MLAT, -85.1° MLON), and Clyde River (CLY, 78.8°
MLAT, 18.1° MLON) to measure the ionospheric convection near the dayside cusp. The three
radars have overlapping field of views (FOVs), enabling a reliable determination of the 2-d
convection velocity. The FOVs cover the ionosphere >75° MLAT, covering the typical location
of the cusp under weak and modest solar wind driving conditions [i.e., *Newell et al.*, 1989] and the
high occurrence region of pulsed ionospheric flows [*Provan and Yeoman,* 1999] with high spatial
resolution. Data from Saskatoon (SAS, 60° MLAT, -43.8° MLON) and Prince George (PGR, 59.6°
MLAT, -64.3° MLON) radars are also used when data are available. The measurements of these
two radars at far range gates can overlap with the cusp. The radar data have a time resolution of 1-
2 min. We focus on observations ±3 h MLT from magnetic noon (approximately 1600-2200 UT).



The satellite footprints should be mapped close to the radar FOVs under the Tsyganenko (T89)
model [*Tsyganenko*, 1989]. Footprints mapped using different Tsyganenko (e.g., T96 or T01
[*Tsyganenko*, 1995, 2002a, 2002b]) models have similar longitudinal locations (difference <100
km), implying the longitudinal uncertainty of mapping to be small. The latitudinal uncertainty can
be inferred by referring to the open-closed field line boundary as estimated using the 150 m/s
spectral width boundary [e.g., *Baker et al.*, 1995, 1997; *Chisham and Freeman*, 2003]. And T89
has given the smallest latitudinal uncertainty for the studied events. We surveyed years 2014-2016
during the months when the satellite apogee was on the dayside, and found 6 such conjunctions.

The ionospheric signature of reconnection includes fast anti-sunward flows moving across the

open-closed field line boundary. We obtain the flow velocity vectors by merging line-of-sight
(LOS) measurements at the radar common FOVs [*Ruohoniemi and Baker*, 1998], and these merged
vectors reflect the true ionospheric convection velocity. However, the radar common FOVs are
hundreds of km wide only, which can be too small to cover the full azimuthal extent of the
reconnection-related flows (which are up to thousands of km wide). We therefore also reconstruct
the velocity field using the Spherical Elementary Current Systems (SECS) method [*Amm et al.*,
2010]. Similar to the works by *Ruohoniemi et al.* [1989] and *Bristow et al.* [2016], the SECS
method reconstructs a divergence-free flow pattern using all LOS velocity data. We refer to these
velocities as SECS velocities. The accuracy of SECS velocities can be validated by comparing to
the LOS measurements and the merged vectors. SECS velocities work best in regions with dense
echo coverage and those around sparse echoes are not reliable and thus are excluded from our
analysis.

The third way of obtaining a velocity field is Spherical Harmonic Fit (SHF). This method uses

the LOS measurements and a statistical convection model to fit the distribution of electrostatic





potential, which is expressed as a sum of spherical harmonic functions [*Ruohoniemi and Baker*,
1998]. The statistical model employed here is *Cousins and Shepherd* [2010]. While this method
may suppress small or meso-scale velocity details, such as, sharp flow gradients or flow vortices,
we compare SHF velocities with the LOS measurements and merged vectors to determine how
well the SHF velocities depict the velocity details.

Among the six events we identified, we present three representative conjunction events in

Sections 3.1-3.3. The time separation of magnetopause crossings by two satellites are 1, 2, and 30
min. While the time separation for the third case is somewhat long, we distinguish the spatial and
temporal effects using the radar data. Although the three events occurred under similar IMF Bz
conditions, the reconnection-related flows in the ionosphere had an azimuthal extent varying from
a few hundred km (Sections 3.1-3.2) to more than a thousand km wide (Section 3.3). This
corresponds to X-lines of a few to >10 Re long, indicating that both spatially patchy (a few Re)
and spatially continuous and extended reconnection (>10 Re) are possible forms of reconnection
at the magnetopause. Interestingly, extended reconnection was found to arise from a spatially
localized patch that spreads azimuthally. Potential effects of IMF Bx and By on the reconnection
extent are discussed in Section 3.4.

3. Observations
3.1. Spatially patchy reconnection active at one satellite only
3.1.1. In-situ satellite measurements

On March 11, 2014, THA and THD made simultaneous measurements of the dayside

magnetopause with a 4.2 Re separation in the Y direction. The IMF condition is displayed in Figure
1a and the IMF was directed southward. The satellite location in the GSM coordinates is displayed





in Figure 1b, and the measurements are presented in Figures 1c-ll. The magnetic field and the ion
velocity components are displayed in the LMN boundary normal coordinate system, where $L$ is
along the outflow direction, $M$ is along the X-line, and $N$ is the current sheet normal. The
coordinate system is obtained from the minimum variance analysis of the magnetic field at each
magnetopause crossing [*Sonnerup and Cahill*, 1967]. Both satellites passed from the
magnetosphere into the magnetosheath, as seen as the sharp changes in the magnetic field, the ion
spectra, and the density (shaded in pink).
As THD crossed the magnetopause boundary layer (1624:47-1625:09 UT), it detected both fluid
and kinetic signatures of reconnection. It observed a rapid, northward-directed plasma jet within
the region where the magnetic field rotated (Figures 1c and 1f). The magnitude of this jet reached
138 km/s at its peak, which was 60% of the predicted speed of a reconnection jet by the Walen
relation (231 km/s, not shown). The angle between the observed and predicted jets was 22°. The
jet velocity was somewhat small (although still sufficiently large that $\Delta V^* > 0.5$) because of the
presence of cold magnetospheric ions seen in Figure 1g [*Phan et al.*, 2013]. Figure 1g suggests
that the magnetosheath ion population had a parallel velocity of ~200 km/s, and the cold
magnetospheric ion population had a parallel velocity near zero. Therefore although the bulk
velocity computed by combining the two populations was considerably slower than the Walen
prediction, the velocity of the magnetosheath population was actually very close to the prediction.
The ion distributions in Figure 1g showed a characteristic D-shaped distribution, consistent with
active reconnection.
THA crossed the magnetopause one minute earlier than THD (1623:29-1624:07 UT). While it
still identified a plasma jet at the magnetopause (Figures 1h and 1k), the jet speed was significantly
smaller than what was predicted for a reconnection jet (97 km/s versus 200 km/s). The observed

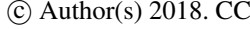



jet was directed 21° away from the prediction. No clear D-shaped distributions have been found
in the ion distributions at the magnetopause (Figure 1l). Reconnection was thus much less active
at THA local time than at THD. This suggests that the X-line of the active reconnection at THD
likely did not extend to THA.

3.1.2 Ground radar measurements

The velocity field of the dayside cusp ionosphere during the satellite measurements is shown in

Figure 2 (the 1-min difference from the satellite magnetopause crossing time is negligible as it was
within the 1-2-min radar resolution). Figure 2a shows the radar LOS measurements, as denoted by
the color tiles, and the merged vectors, as denoted by the arrows. The colors of the arrows indicate
the merged velocity magnitudes, and the colors of the tiles indicate the LOS speeds that direct anti-
sunward (those project to the sunward direction appear as black). Fast (red) and anti-sunward flows
are the feature of our interest. One channel of such flow can be identified in the pre-noon sector,
which had a speed of ~700 m/s and was directed poleward and westward. The velocity
vectors >~80° MLAT were directed roughly parallel to the RKN radar beams, and therefore the
RKN LOS measurements represent the primary component of the flow. The flow crossed the open-
closed field line boundary, which stayed quasi-steadily at 77° MLAT based on the spectral width
(Figure 2d discussed below). This flow thus meets the criteria of being an ionospheric signature
of magnetopause reconnection.

The flow had a limited azimuthal extent. The extent is determined at half of the maximum flow

speed, which was ~400 m/s. Figure 2e discussed below shows a more quantitative estimate of the
extent. In Figure 2a, we mark the eastern boundary with the dashed magenta line, across which the
velocity vectors at 79°-83° MLAT dropped from red/orange to green color. Those green vectors,



different from the red vectors, were directed mainly westward roughly in parallel to the CLY radar
beams. They had a small poleward velocity component, or even an equatorward component, up to
2 h in MLT past magnetic noon as seen from the dark blue and the black LOS measurements from
RKN and SAS. They hence were the slow background convection outside the fast anti-sunward
flow. The western boundary of the flow had extended beyond the RKN FOV. But it did not extend
more than 1.5 h in MLT beyond because the INV echoes there showed weakly poleward and
equatorward LOS speeds across the open-closed field line boundary.

It is possible to infer the location of the flow western boundary more definitively from the SECS

velocities than the LOS measurements. Figure 2b shows the SECS velocities, denoted by the
arrows. The SECS velocities reasonably reproduced the spatial structure of the flow channel seen
in Figure 2a. The flow western boundary was marked by the dashed magenta line, across which
the flow speed dropped and the flow direction reversed. The equatorward-directed flows are
interpreted as the return flow of the poleward flows, as sketched in *Southwood* [1987] and *Oksavik*
*et al.* [2004].

The velocity field reconstructed using the SHF velocities is shown in Figure 2c (obtained

through the Radar Software Toolkit (http://superdarn.thayer.dartmouth.edu/software.html)). The
SHF velocities also exhibit a channel of fast poleward and westward directed flow, which was
similar to the flow channel in Figure 2b. The flow western and eastern boundaries were again
marked by the dashed magenta lines (using the same ~400-m/s threshold as above), across which
the SHF velocities dropped from orange to green/blue.

We can test the reliability of the identified flow extent by referring to the extent of the cusp.

Evidence has shown that the longitudinal extent of the cusp correlates with the extent of PMAFs
[*Moen et al.*, 2000] and of poleward flows across the open-closed field line boundary [*Pinnock*



*and Rodger*, 2001]. Figure 2d shows spectral width measurements and the cusp can be identified
as a region of high spectral widths (red color) as circled in the red contour. The cusp was located
at the western half of the RKN FOV and its eastern edge corresponded to a drop of the spectral
widths from red to green color. The western edge extended beyond the RKN FOV and the
extension was partially captured by the PGR echoes (marked by the dashed line as the backscatters
there had gaps in space and were sporadic in time). But it ended around the low spectral widths of
the CLY backscatters eastward of the INV FOV. The location and the extent of the cusp therefore
support the location and the extent of the anti-sunward flow.

The limited extent of the flow did not vary much in time, as suggested by the time series plot in

Figure 2e. Figure 2e presents the RKN LOS measurements along 80° MLAT as functions of
magnetic longitude (MLON) and time. Similar to the snapshots, the color represents LOS speeds
that project to the anti-sunward direction, and the flow of our interest appears as a region of red
color. The time when THA and THD crossed the magnetopause was marked by the red arrows.
The fact that the flow channel stayed quasi-steady during the satellite conjunction period suggests
that the satellite measurements in Section 3.1.1 reflect the spatial distribution, rather than the
temporal variation, of reconnection.

Figures 2a-c all observed a channel of fast anti-sunward flow in the pre-noon sector of the high

latitude ionosphere, and the channel had a limited azimuthal extent. If the flow corresponded to a
magnetopause reconnection, the X-line is expected to be located in the GSM-Y < 0 regime and
spans over a limited local time range. This is consistent with the THEMIS satellite observation in
Section 3.1.1, where THD at Y = -2.0 Re detected clear reconnection signatures, while THA at Y
= 2.2 Re did not. In fact, if we project the satellite location to the ionosphere through field line
tracing under the T89 model, THD was positioned close to the flow eastern boundary, while THA



was far away (Figures 2a-c).

The radar observations thus provide critical information to interpret the in-situ reconnection

extent. The X-line detected by THD did not extend duskward passing through the subsolar point
of the magnetosphere; instead it extended dawnward towards the dawnside magnetopause. Note
that the observations presented here do not rule out existence of other X-lines along the
magnetopause, as there might exist other fast anti-sunward flows outside the radar FOVs. But those
X-lines are not the focus of this study. It should also be noted that the determined flow extent is
based on half of the maximum flow speed, which allows weak anti-sunward flows to extend
beyond the flow boundaries. The weak flows are expected to correspond to weak background
reconnection at the magnetopause. In fact, THA had detected weak reconnection signatures (i.e.
weak plasma jets) 4-Re eastward of the active reconnection signatures at THD as found in Section
3.1.1, and this may agree with the weak ionospheric convection (green vectors in Figures 2a-c)
eastward of the flow channel.

We quantitatively determine the flow extent in Figure 2f. Figure 2f shows the RKN LOS

velocity profile along 80° MLAT at 1624 UT (the same time as Figures 2a-c) as a function of
magnetic longitude and distance from 0° MLON. As mentioned above, the RKN LOS
measurements captured the flow major component and thus approximated to the true 2-d velocities.
Also shown is the SECS velocity profile. Here we only show the northward component of the
SECS velocity as this component represents reconnecting flows across an azimuthally-aligned
open-closed field line boundary. We quantify the flow azimuthal extent as the full-width-at-half-
maximum (FWHM) of the velocity profile. But the LOS measurements only captured part of the
flow channel, and could only reveal the half-width-at-half-maximum (HWHM) on one side of the
velocity profile. The HWHM was 15° in MLON and 300 km at an altitude of 250 km. The SECS



velocities covered the entire flow channel, and can be used to determine the FWHM. The FWHM
was 26° in MLON and 520 km.

To infer the X-line extent at the magnetopause, we project the flow width in the ionosphere to

the equatorial plane. This is done by mapping a pair of ionospheric locations that are azimuthally
separated around the THD footprint to the equatorial plane. The ratio of the pair separation in the
equatorial plane to that in the ionosphere gives a mapping factor. The mapping factor under T89
is 55, and this suggests the X-line local time extent to be ~4 Re.

3.2. Spatially patchy reconnection active at both satellites
3.2.1. In-situ satellite measurements

On April 19, 2015, under a southward IMF (Figure 3a), THA and THE crossed the

magnetopause nearly simultaneously (<2 min lag) with a 0.5 Re separation in Y (Figure 3b). They
passed from the magnetosheath into the magnetosphere. Both satellites observed jets in the $V_L$
component at the magnetopause. The jet at THA at ~1828:05 UT had a speed of 84% of and an
angle within ~15° from the Walen prediction. The jet at THE at ~1826:25 UT had a speed of 95%
of and an angle of ~29° from the Walen prediction. The ion distributions at THA and THE exhibit
clear D-shaped distributions, indicative of active reconnection at these two local times.

Section 3.2.2. Ground radar measurements

During the satellite measurements, the radars observed a channel of fast anti-sunward flow

around magnetic noon (Figures 4a-c). The flow crossed the open-closed field line boundary at 77°
MLAT, and qualifies for an ionospheric signature of magnetopause reconnection. The flow
direction was nearly parallel to the RKN radar beams, and therefore the RKN LOS measurements





in Figure 4a approximated to the 2-d flow speed. The flow eastern boundary can be identified as
where the velocity dropped from red/orange to blue (dashed magenta line). Determining the flow
western boundary requires more measurements of the background convection velocity, which is
beyond the RKN FOV. But we again infer that the western boundary did not extend more than 1.5
h westward beyond the RKN FOV because the PGR and INV echoes there showed weakly
poleward and equatorward LOS speeds around the open-closed field line boundary. The CLY radar
data further indicated that the anti-sunward flow had started to rotate westward immediately
beyond the RKN FOV. This is because the CLY LOS velocities measured between the RKN and
INV radar FOVs were larger for more east-west oriented beams (appearing as yellow color) than
for more north-south oriented beams (green color). The rotation likely corresponds to the vortex
at the flow western boundary as sketched in *Oksavik et al.* [2004].

The more precise location of the western boundary can be retrieved from the SECS velocities

in Figure 4b and the SHF velocities in Figure 4c. The SECS velocities present a flow channel very
similar to that in Figure 4a, while the flow channel in the SHF velocities was more azimuthally-
aligned than in Figures 4a-b.

The determined flow extent agrees with the extent of the cusp in Figure 4d. The high spectral

widths associated with the cusp were located at the western half of the RKN FOV. They extended
westward beyond the RKN FOV into CLY far range gates, where they dropped from red to green
color. This is consistent with the inferred location and extent of the anti-sunward flow.

The flow of our interest just emerged from a weak background at the time when the THEMIS

satellites crossed the magnetopause (Figure 4e). This implies that the related reconnection just
initiated at the studied local time. The flow and the reconnection remained with a roughly steady
and localized extent after formation. We quantify the HWHM of the flow using the RKN LOS

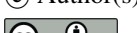



velocity profile at 0830 UT (Figure 4f), and the HWHM was 10° MLON and 220 km. The FWHM
is determined using the SECS velocities, and the FWHM was 26° MLON and 572 km. Such an
FWHM corresponds to ~5 Re in the equatorial plane.

The fact that the fast anti-sunward flow had a limited azimuthal extent around magnetic noon

implies that the corresponding magnetopause X-line should span over a limited local time range
around the noon. This is consistent with the THEMIS satellite observation in Section 3.2.1, where
reconnection was active at Y = 0.7 (THA) and 0.2 Re (THE). Projecting THA and THE locations
to the ionosphere reveals that both satellite footprints were located within the flow longitudes.
Therefore the reconnection at the two satellites was part of the same X-line around the subsolar
point of the magnetopause. (The THE footprint was equatorward of THA because the X location
of THE was closer to the Earth than THA. The magnetopause was expanding and it swept across
THE and then THA.) The X-line further extended azimuthally beyond the two satellite locations,
reaching a full length of ~5 Re.

3.3. Spatially continuous and extended reconnection active at both satellites
3.3.1. In-situ satellite measurements

On Apr 29, 2015, under a prolonged and steady southward IMF, THA and THE crossed the

magnetopause successively with a time separation of ~30 min. The locations of the crossings were
separated by 0.1-0.2 Re in the Y direction (Figure 5b). The satellites passed from the
magnetosphere into the magnetosheath, and the magnetic field data suggest that the satellites
crossed the current layer multiple times before completely entering the magnetosheath. We
therefore only display the magnetic field and the plasma velocity in the GSM coordinates. Both
satellites detected multiple flow jets, all agreeing with the Walen prediction with $\Delta V^* > 0.5$. For





example, the jet at 1849-1850 UT measured by THA had a speed with 80% of and angle with 9°
from the Walen prediction, and the jet at 1920-1922 UT by THE had a speed with 83% of and an
angle with 1° from the Walen prediction. The ion distributions at THA and THE exhibit clear D-
shaped distributions. Such observations suggest that reconnection was active at the THA and THE
local times.

3.3.2. Ground radar measurements
In the ionosphere, the radars detected a fast anti-sunward flow as an ionospheric signature of
the magnetopause reconnection (Figures 6a-c). The flow velocity here had a large component
along the looking directions of the INV and CLY radars, and we therefore focus on the LOS
measurements of these two radars. The flow had a broad azimuthal extent, as delineated by the
dashed magenta lines (Figure 6a). A similar flow distribution is found in the SECS velocities
(Figure 6b), and the SHF velocities (Figure 6c). Corresponding to the broad extent of the flow, the
cusp had a broad extent (Figure 6d). The cusp continuously spanned across the INV and RKN
FOVs and its western and eastern edges coincided with the western and eastern boundaries of the
flow, supporting our delineation of the flow extent.
The wide flow channel in the ionosphere implies that the corresponding magnetopause X-line
should be wide. Based on the flow distribution, we infer that much of the X-line should be located
on the pre-noon sector, except that the eastern edge can extend across the magnetic noon meridian
to the early post-noon sector. This inference is again consistent with the inference from the THA
and THE measurements that the reconnection extended at least over the satellite separation (Y = -
0.2 (THA) and 0 Re (THE)). Note, however, that the distance between THA and THE only covered
<2% of the X-line extent determined from the ionosphere flow. While the satellite configuration



and measurements here were similar to those in Section 3.2, the extent of reconnection was
fundamentally different. This suggests that it is difficult to obtain a reliable estimate of the
reconnection extent without the support of 2-d measurements and that satellites alone also cannot
differentiate spatially extended reconnection from spatially patchy reconnection.

The flow temporal evolution is shown in Figures 6e-f, where the velocities are based on the LOS

measurements from the CLY (Figure 6e) and INV (Figure 6f) radars. The velocities >-18° MLON
are not useful and are shaded in grey. These measurements were from short range gates of the CLY
radar, where the convection velocity is underestimated as the Doppler velocity is limited below
the ion acoustic speed (~400 m/s) [*Haldoupis*, 1989; *Koustov et al*., 2005]. An overall wide flow
channel is seen between ~-90° and -30° MLON for most of the studied time period, and in
particular the flow azimuthal extent were nearly identical at the instances when THA and THE
observed the reconnection. But between the two satellite observations, the flow experienced an
interesting variation. The velocity at -74°−-30° MLON dropped by 100-200 m/s during 1900-1910
UT, while the velocity at -88°-74° MLON did not change substantially. The velocity enhanced
again from 1910 UT. The enhancement first occurred at ~-60°−-40° MLON and then spread
azimuthally towards east and west. The enhancement spread by 18° over 14 min at its eastern end
(marked by the dashed magenta line), suggesting a spreading speed of 429 m/s. The spreading at
the western end soon merged with the velocity enhancement at -88°-74° MLON, but a rough
estimate suggests a speed of 444 km/s. It should be noted that the all three components of the IMF
stayed steady for an extended time (Figure 7, discussed below in Section 3.4), and thus the
evolution of the flow/reconnection was unlikely to be externally driven.

This sequence of changes gives an important implication that the extended X-line was a result

of spreading of an initially patchy X-line. If we map the spreading in the ionosphere to the



485 magnetopause, the spreading occurred bi-directionally and at a speed of 24 km/s in each direction

486 based on field-line mapping under the T89 model (the mapping factor was 55). The spreading

487 process persisted for 10-20 min. Such an observation is similar to what has recently been reported

488 by *Zou et al.* [2018], where the X-lines also spread bi-directionally at a speed of a few tens of km/s.

489 However, the spreading in *Zou et al.* [2018] occurs following a southward turning of the IMF,

490 while the spreading here occurred without IMF variations. The mechanism of spreading is

491 explained either as motion of the current carriers of the reconnecting current sheet or as

492 propagation of the Alfven waves along the guide field [*Huba and Rudakov,* 2002; *Shay et al.* 2003;

493 *Lapenta et al.*, 2006; *Nakamura et al.*, 2012; *Jain et al.*, 2013].

494  It should be noted that X-line spreading can be a common process of reconnection that is not

495 limited to extended X-lines. A careful examination of Figure 4d suggests that spreading may have

496 also occurred for the patchy X-line (the eastern limit of the red/orange region spread from -36° to

497 -29° MLON during 1828-1832 UT). The two X-lines spread at a similarly speed, but duration of

498 the spreading process was two to three times longer in the extended than the patchy reconnection

499 events.

500  Figures 6g-h quantify the FWHM of the fast anti-sunward flow around the time when THA and

501 THE measured active reconnection. The width can be obtained based on the LOS measurements,

502 where we determine the HWHMs of the flow in the INV and CLY FOVs separately and add them

503 together as the FWHM. The FWHM was 63° MLON and 1260 km when THA measured the

504 reconnection, and was 62° MLON and 1240 km when THE measured the reconnection. This

505 corresponds to an X-line length of ~11 Re. Note that the determination of the HWHM inside the

506 CLY FOV has taken into account a background convection of ~400 m/s. The background came

507 from those plasmas moving azimuthally along the open-closed field line boundary but not crossing



the boundary. The width can also be obtained based on the SECS measurements, which was 64°
MLON and 1280 km when THA measured the reconnection, and 60° MLON and 1200 km when
THE measured the reconnection. This is very close to the values derived from the LOS
measurements.

3.4. IMF and solar wind conditions for spatially patchy and extended reconnection

The above events definitely show that the local time extent of magnetopause reconnection X-

lines can vary from a few to >10 Re. Here we investigate whether and how the extent may depend
on the upstream driving conditions. Figure 7 presents the IMF, the solar wind velocity, and the
solar wind pressure taken from the OMNI data for the three events. The red vertical lines mark the
times when the reconnection was measured. The three events occurred under similar IMF field
strengths (5-6 nT), similar IMF Bz components (-3 nT), and similar solar wind velocities (300-400
km/s) and dynamic pressures (1-2 nPa), implying that the different X-line extents were unlikely
due to these parameters. This is different from *Milan et al.* [2016], who identified the solar wind
velocity as the controlling factor of reconnection extent. However, *Milan et al.* [2016] studied
reconnection under very strong IMF driving conditions when |B| ~15 nT, while our events occurred
under a more typical moderate driving (|B| ~5-6 nT).

The patchy X-line events had an IMF Bx of a larger magnitude than the extended reconnection

event did (4 vs. 0 nT). The patchy X-line events also had an IMF By component of a smaller
magnitude (2 vs. 5 nT), and with more variability on time scales of tens of minutes, than the
extended X-line event. The IMF Bx and By components are known to modify the magnetic shear
across the magnetopause and to affect the occurrence location of reconnection. The steady IMF
condition may allow X-lines to spread across local times unperturbedly, eventually reaching a wide



extent. Thus the X-line extent may depend on the IMF orientation and steadiness, although whether
and how they influence the extent needs to be further explored.

4. Summary

We carefully investigate the local time extent of magnetopause reconnection X-lines by

comparing the measurements of two THEMIS satellites and three ground radars. The radars
identify signatures of reconnection as fast ionospheric flows moving anti-sunward across the open-
closed field line boundary. When reconnection is active at only one of the two satellite locations,
only the ionosphere conjugate to this spacecraft shows a channel of fast anti-sunward flow. When
reconnection is active at both spacecraft and the spacecraft are separated by <1 Re, the ionosphere
conjugate to both spacecraft shows a channel of fast anti-sunward flow. The fact that the satellite
locations are mapped to the same flow channel suggests that the X-line is continuous between the
two satellites, and that it is appropriate to take the satellite separation as a lower limit estimate of
the X-line extent. Whether the X-line can still be regarded as continuous when the satellites are
separated by a few or > 10 Re is questionable, and needs to be examined using conjunctions with
a larger satellite separation than what have been presented here.

The X-line extent is measured as the extent of the ionospheric flow. In the three conjunction

events, the flows have an extent of 520, 572, and 1260 km in the ionosphere, which corresponds
to ~4, 5, and 11 Re at the magnetopause (under the T89 model) in the local time direction. This
provides strong observational evidence that magnetopause reconnection can occur over a wide
range of extents, from spatially patchy (a few Re) to spatially continuous and extended (>10 Re).
Interestingly, the extended reconnection is seen to initiate from a patchy reconnection, where the
X-line grows by spreading across local time. The speed of spreading is 50 km/s summing the

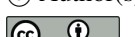



westward and eastward spreading motion, and the spreading process persists for 10-20 min.
The X-line extent may be affected by the IMF orientation and steadiness, although the
mechanism is not clearly known. For the modest solar wind driving conditions studied here, the
extended X-line occurs under a smaller IMF Bx component, and a larger and steadier IMF By
component than the patchy X-line. The IMF strength, the Bz component, and the solar wind
velocity and pressure are about the same for the extended and the patchy X-lines. Reconnection
can vary with time, even under steady IMF driving conditions.

**Acknowledgments.** This research was supported by the NASA Living With a Star Jack Eddy
Postdoctoral Fellowship Program, administered by UCAR's Cooperative Programs for the
Advancement of Earth System Science (CPAESS), NASA grant NNX15AI62G, NSF grants PLR-
1341359 and AGS-1451911, and AFOSR FA9550-15-1-0179 and FA9559-16-1-0364. The
THEMIS mission is supported by NASA contract NAS5-02099. SuperDARN is a collection of
radars funded by national scientific funding agencies. SuperDARN Canada is supported by the
Canada Foundation for Innovation, the Canadian Space Agency, and the Province of
Saskatchewan. We thank Tomoaki Hori for useful discussion on the SECS technique. Data
products of the SuperDARN, THEMIS, and OMNI are available at http://vt.superdarn.org/,
http://themis.ssl.berkeley.edu/index.shtml, and GSFC/SPDF OMNIWeb website.

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



Figure 1. Measurements from THD and THA during their nearly simultaneous crossings of the
magnetopause on March 11, 2014. Figure 1a: OMNI IMF condition. Figure 1b: THD and THA
locations projected to the GSM X-Y plane. The dashed curve marks the magnetopause and the
dotted curve marks the bow shock. Figures 1c-f: THD measured magnetic field (0.25 s resolution),
ion energy flux (3 s), ion density (3 s), and ion velocity (3 s). The ion measurements were taken
from ground ESA moments. The magnetic field and the ion velocity components are displayed in
the LMN boundary normal coordinate system. The magnetopause crossing is shaded in pink.
Figure 1g: THD ion distribution function on the bulk velocity-magnetic field plane. The small
black line indicates the direction and the bulk velocity of the distributions. Figures 1h-l: THA
measurements in the same format as in Figures 1c-g.

Figure 2. Ionospheric velocity field at the cusp when the THEMIS satellites crossed the
magnetopause on March 11, 2014. Figure 2a: SuperDARN LOS speeds (color tiles) and merged
velocity vectors (color arrows) in the Altitude adjusted corrected geomagnetic (AACGM)
coordinates. The FOVs of the RKN, INV, and CLY radars are outlined with the black dashed lines.
The colors of the tiles indicate the LOS speeds away from the radar. The colors and the lengths of
the arrows indicate the merged velocity magnitudes and the arrow directions indicate the velocity
directions. Red and anti-sunward directed flows are the ionospheric signature of magnetopause
reconnection. The dashed magenta lines mark the flow western and eastern boundaries. The
satellite footprints under the T89 are shown as the THD and THA marker. Figure 2b: Similar to
Figure 2a but showing SECS velocity vectors (color arrows). Figure 2c: Similar to Figure 2a but
showing SHF velocity vectors (color arrows). Figure 2d: SuperDARN spectral width
measurements (color tiles). The red contour marks the cusp. Figure 2e: Time evolution of RKN





LOS velocities along 80° MLAT. The velocities are color coded in the same way as Figure 2a.
Figure 2f: Longitudinal profile of convection velocities along 80° MLAT at 1622 UT. The profiles
is also shown as a function of the distance measured azimuthally from 0° MLON. The profile in
black is based on the RKN LOS measurements, from which the HWHM is determined and marked
by the black arrow. The profile in red is based on the northward components of the SECS velocities,
from which the FWHM is determined and marked by the red arrow. The dotted black and red
vertical lines are the drop lines of the HWHM and FWHM, respectively.

Figure 3. Measurements from THA and THE during their nearly simultaneous crossings of the
magnetopause on Apr 19, 2015. The figure format is similar to Figure 1.

Figure 4. Ionospheric velocity field at the cusp when the THEMIS satellites crossed the
magnetopause on Apr 19, 2015. The figure format is similar to Figure 2. The velocity time
evolution in Figure 4e and the velocity profile in Figure 4f are taken along 79 °MLAT.

Figure 5. Measurements from THA and THE during their crossings of the magnetopause on Apr
29, 2015. The figure format is similar to Figure 1, but the magnetic field and plasma velocities are
displayed in the GSM coordinates.

Figure 6. Figures 6a-d: Ionospheric velocity field at the cusp when the THEMIS satellites crossed
the magnetopause on Apr 29, 2015. The figure format is similar to Figures 2a-d except that in
Figure 6a the color of the CLY color tiles represent LOS speeds towards the radar as here LOS
speeds towards the CLY radar project to the anti-sunward direction. Figures 6e-f: Time evolution

Ann. Geophys. Discuss., https://doi.org/10.5194/angeo-2018-63


of LOS velocities along 80° MLAT from the INV and CLY radars. The velocity measurements in
the shaded region are backscatters from the E-region ionosphere and thus underestimate the
convection speed. The flow channel spread azimuthally before reaching an extended extent, and
the time-dependent locations of its western and eastern boundaries are marked by the dashed
magenta lines. Figures 6g-h: Longitudinal profiles of the LOS and the poleward SECS velocities
along 80° MLAT when THA and THE observed reconnection.

Figure 7. Comparison of the IMF and solar wind driving conditions between the reconnection
events on March 11, 2014, Apr 19, 2015, and Apr 29, 2015. From top to bottom: IMF in GSM
coordinates, solar wind speed, and solar wind dynamic pressure. The red vertical lines mark the
times of the satellite-ground conjunction.





Figure 1.



Figure 2.



Figure 3,






Figure 4,

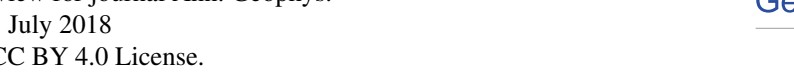



Figure 5.



Figure 6.






Figure 7.

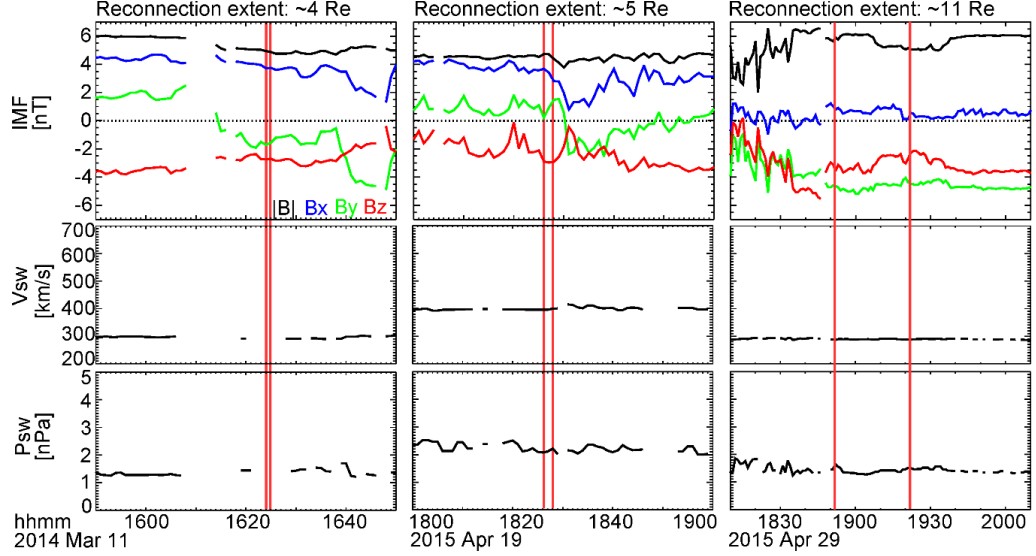












