# Peer review of "Local time extent of magnetopause reconnection using space-ground coordination"

_Annales Geophysicae, 2018_

## Referee Comment (RC1) · Anonymous Referee #1 · 15 Aug 2018

Local time extent of magnetopause reconnection X-lines using space-ground coordination

Author: Ying Zou

The authors discuss conjugate observations of two THEMIS satellites crossing the magnetopause in short succession, with ground based radar observations to determine the lengths of a dayside reconnection line at the magnetopause. The methodology seems to be interesting and the paper is well written with a very good introduction to the general problem. However I have some issues with the current data analysis and the event selection that are significant enough to not recommend publication at this time.

[Figure]

General Point: As the authors admit, their determined length of the X-line will be limited to the longitudinal coverage of the radars. This unavoidable limitation will always significantly influence their conclusion about the length of the "actual" X-line, which could be considerably longer, and will prevent them from ever finding a global answer. That will seriously limit the usefulness of the methodology, thought generally its an interesting approach.

About the introduction: There are several significant publications using IMAGE/FUV observations. This mission had the ability to observe emissions from precipitating (cusp) ions over the entire polar region at once and was therefore not limited like the radar coverage in the present manuscript. Studies using these data have shown evidence that during southward IMF conditions the entire dayside is open leading to very long dayside reconnection lines. So, based on these results the length of the X-line is not the driving question. In additions, decades of cusp observations in all local time sectors show precipitating ions. X-lines in general seem to be very long.

Cusp observations have shown that a substantial part of reconnection is dominated by pulsed reconnection [Lockwood et al.,…...]. The question is therefore – is the long X-line pulsing as "One" or are individual longitudinal sections have their own pulsation frequency? That should lead to scenarios presented in this manuscript, sections of X-lines that are active next to sections of X-lines temporarily inactive. This is how I would interpret the observations in the manuscript. Therefore the conclusion would not be about the length of the X-line since that would be masked by the temporal nature of the reconnection process, which might lead to misleading results.

In any case, I was surprised that there was no reference to this rather ground breaking IMAGE observations anywhere. These observations [e.g., Fuselier et al., 2002] should be added in the introduction and properly described.

Specific Points:

Line 188: the D-shaped distribution do not persist into the ionosphere due to the conservation of the first adiabatic invariant. The D shape changes into a Crescent shape as soon as the ambient B field increases, which it definitely will in the cusps. This has been observed in the cusp regions for decades. This effect is so pronounced that it can be even used directly at the magnetopause. The "bending over" of the D-shape distribution observed during magnetopause crossings has been used in a recent study by Broll et al. (2017) (JGR) to determine the distance to the X-line from the MMS satellites and infer the Xline location.

Cusp Steps have nothing to do with D-shape distributions. Cusp steps are the result of changes in the reconnection rate at the magnetopause or caused by spatially separated X-lines. Cusp-steps have been discussed in great detail by Lockwood and Smith in the 90ties as manifestation of pulsed reconnection leading to the pulsed reconnection model and by e.g., Onsager et al [1995] or Trattner et al. [2002] as spatially separated X-lines.

The authors use patchy reconnection also in the case of spatially separated X-line or partial X-lines. This will be a source of confusion for colleagues not too familiar with the subject. Patchy reconnection usually describes pulsed reconnection – temporal changes in reconnection. While the authors do a reasonable good job in trying to keep the temporal and spatial regimes apart I would recommend to revisit that issue throughout the paper.

Figure 2: The symbol for Th-D is completely invisible – if it wasn't for Figure 4 I would not have realized that there are indeed two separate magnetic foot points in that plot. Chose a different more prominent color.

Figure 2: it is mentioned in line 209 – the satellite foot points should map close to the radars FOV. I would recommend that the authors look for events where the satellite foot points are actually in the FOV of the radars to make absolutely sure that these observations are linked. Throughout the paper but especially in Figure 2 I do not have the impression that this is the case which makes the data analysis rather questionable.

Therefore I fail to see how the observed D-shape distributions at the magnetopause are connected with particular flow channels which is the essential part of the study.

The authors also mark the cusp foot point in the radar images. Discussing again the events in figure 2, Th-D clearly saw an ion jet. It therefore observed reconnection at the magnetopause and was on a newly opened field line. The D shape distribution, while looking a bid crooked compared to the other D-shape distributions in the manuscript, travels along the magnetic field. The magnetic field, at that time the distribution was observed, was still northward. Therefore the satellite was in the LLBL and the ions move toward the northern cusp where the radar observations observe flow channels. All open magnetopause field lines map into the cusps. So the Th-D magnetic foot point, were the D-distribution was observed, should be in that region marked as cusp in figure 2d. It is not, its not even in the FOV for the radar.

Line 338: One of the open questions in magnetic reconnection is still how the reconnection rate develops along the length of the X-lines. Since decades of research showed that pulsed reconnection is a rather significant process, it is conceivable that individual sections along a "long" X-line pulse at different frequencies. I therefore would expect that it is very likely that magnetopause crossings by multiple satellites show active and temporarily inactive sections along an X-line. This is not prove that a dayside X-line is short. The interpretation of the authors that this event is a spatially restricted X-line based on flow channels at very different latitudes is not convincing, especially since the satellite observations are outside the flow channels for which observations exist.

I also want to stress that in the pulsed reconnection model, field lines that were opened before reconnection briefly stopped, are convecting and provide a continuous transfer of magnetosheath plasma into the magnetosphere. That should certainly influence your radar observations. It is unlikely that the ionosphere would respond that quickly to short changes in the reconnection rate. The magnetosphere is generally rather slow in its response to outside changes. That will make linking ionospheric flow channels to magnetopause observations rather challenging.
Radar observations of ionospheric convection, direction and velocities, are often used to estimate global convection pattern in the polar ionosphere using various models. These "convection cells" could be overlayed in the radar plots to make a connection between the satellite magnetic foot points outside the radar FOV and the radar data. Depending on how these global convection cells look like they might provide a more convincing picture that these observations are actually linked.
* * *

---

## Referee Comment (RC2) · Anonymous Referee #2 · 24 Aug 2018

This paper is concerned with estimating the extent of reconnection X-lines on the Earth's magnetopause, with an overall aim of measuring, and understanding spatial and temporal variability in magnetic reconnection. For studies of this type, conjugate observations combining spacecraft and ground-based measurements can be very important. There are some aspects of reconnection (such as the localised plasma physics) that can only be measured by in-situ spacecraft. There are also some aspects (such as the macrophysics of the process) that can only be measured by instruments that provide a wider view, such as auroral imagers or ground-based radars. However, the local time extent of reconnection regions can only be determined unambiguously using ionospheric measurements (in the absence of a massive armada of spacecraft). Similarly, the amount of flux transfer occurring during reconnection can only be determined unambiguously using ionospheric measurements. And consequently, the patchy (spatial variation) and bursty (temporal variation) of reconnection can only be unambiguously studied using ionospheric measurements.

To measure the extent of reconnection from ionospheric measurements (which can then be mapped back to the magnetopause) first requires the identification of the ionospheric footprint of the open-closed magnetic field line boundary (OCB). The regions where the ionospheric plasma flow crosses this boundary (in the frame of the boundary – which is typically in motion itself) map to the regions on the magnetopause where reconnection is occurring. Although the text shows that the authors appear to appreciate this, they do not analyse their ionospheric data in this way.

Consequently, I have some major issues with the introductory text and the radar data analysis and presentation. The authors need to address these major points before the paper can be reviewed properly.

(1) Some of the background referencing is misdirected and inadequate:

The referencing of spacecraft observations associated with reconnection (extending from lines 95 to 117) starts with the phrase – 'The extent of reconnection X-lines has been observationally determined based on fortuitous satellite conjunctions. . .'. This is not true. Even if the word 'determined' was changed to 'estimated' it would still be a stretch of the truth. The 'extent of reconnection X-lines' cannot be unambiguously determined (or even estimated) from spacecraft observations. Interpretations of multiple spacecraft observations still have to make the assumption that the X-line is continuous between spacecraft, or that it is not continuous between spacecraft. X-lines may also continue longitudinally outside of the view of the spacecraft. All that multiple spacecraft measurements can do (given that the assumptions made are correct) is provide upper or lower limits on the X-line extent.

The referencing of ionospheric observations associated with reconnection (extending from lines 118 to 141) concentrates on those related mainly to local (often single

radar) measurements of fast anti-sunward flows observed by radar (such as pulsed ionospheric flows [PIFs]) and their auroral counterpart (poleward-moving auroral forms [PMAFs]). These typically occur within the polar cap, and not necessarily at the ionospheric footprint of the OCB. Although all these observations are of phenomena that are consequences of reconnection, and which provide important information about the patchy and bursty nature of reconnection (and links to FTEs, etc.), they don't allow the unambiguous estimation of the extent of the X-line. Hence, many of these references are actually superfluous to the paper.

As mentioned above, to measure the extent of the reconnection X-line in the ionosphere requires the identification of the footprint of the OCB and the region for which there is plasma flow across it. (Although, similar caveats to the spacecraft observations also exist if there is not complete longitudinal coverage covering the whole ionospheric projection of the X-line.) There are a large number of papers that have studied and measured reconnection in this way that are not mentioned in the introduction of the present paper. A significant reference that reviews most of the work in this area, as well as outlining the techniques required to make these measurements, is Chisham et al. (2008) – Remote sensing of the spatial and temporal structure of magnetopause and magnetotail reconnection from the ionosphere – Rev. Geophys., 46, RG1004. Other papers that have measured the extent of the reconnection X-line using these methods include; (i) Pinnock et al. (2003) – The location and rate of dayside reconnection during an interval of southward interplanetary magnetic field – Ann. Geophys., 21, 1467-1482, which studied the same event that was observed in Equator-S data by Phan et al. (2000). They estimated the length of the reconnection X-line on the dayside magnetopause at this time to be ∼38 Re based on the 10 hours of local time that flow was observed crossing the OCB in the ionosphere. (ii) Chisham et al. (2004) – Measuring the dayside reconnection rate during an interval of due northward interplanetary magnetic field – Ann. Geophys., 22, 4243-4258, which measured the X-line extent of lobe reconnection during northward IMF to be ∼6-11 Re.

[Figure]

(2) Identification of the extent of the reconnection region from fast ionospheric flows is flawed:

Lines 52-54 state – 'The extent has also been inferred by radars as fast ionospheric flows moving anti-sunward across the open-closed field line boundary, but whether a particular ionospheric flow results from reconnection needs to be confirmed.' Firstly, the measured flows do not need to be fast. The fast flows highlighted in the paper are obviously driven by reconnection but these are predominantly polar cap flows (relating to the newly-opened flux tubes moving over the polar regions towards the nightside), not flows at and across the OCB. Any flow across the OCB, whether fast or slow, implies that reconnection has occurred, as closed flux has been converted to open flux. By the same argument, if flow across the OCB is measured, spacecraft measurements are not required to prove that this flow is a result of reconnection (hence I disagree with the statement on lines 132-135).

Lines 198-206 detail the SuperDARN radars used in the study. What I do not understand is why the authors restricted their study to only a few of the northern hemisphere radars when there is a much wider network of northern hemisphere SuperDARN radars that would provide a much greater longitudinal coverage? Larger coverage provides a much better global picture of the ionospheric convection and hence the reconnection-driven flows across the OCB.

Lines 297-298 state – 'The extent is determined at half of the maximum flow speed, which was ∼400 m/s'. Why? There is still flow across the boundary outside this region that results from reconnection. Consequently, the dashed magenta lines in figures 2, 4, and 6 mean nothing, except to nicely frame the fast poleward flows into the polar cap. In a similar vein, lines 366-367 state 'We quantify the flow azimuthal extent as the full-width-at-half-maximum (FWHM) of the velocity profile'. Why? Any poleward flow (across the OCB) represents the creation of newly reconnected flux. In all 3 examples there are significant poleward flows east of the dashed magenta lines.

In figures 2e and 2f the flow extent is 'quantitatively determined' using measurements at 80 degrees latitude. Why use the flows at this latitude to determine the longitudinal extent when they are well within the polar cap? These are not the same as the flows at the OCB latitude, and hence they do not show the longitudinal extent of reconnection. Hence, they cannot be reliably used to estimate the length of the X-line.

(3) The open-closed field line boundary (OCB) in the ionosphere is insufficiently determined:

Lines 390-391 state 'The flow crossed the open-closed field line boundary at 77 degrees MLT...'. The determination of the OCB location is not clearly outlined anywhere or displayed clearly on the figures. Indeed, the OCB location in figures 2, 4, and 6 is never sufficiently determined (or visually presented) so it is impossible to know what the longitudinal extent of flows across the boundary is. The boundary is vaguely discussed as being the equatorward edge of the cusp, which is identified in these figures as being co-located with regions of high Doppler spectral width. (In actuality, comparing figures 2c and 2d, the poleward flow at the equatorward edge of the cusp is slower than that within the polar cap, and most likely extends over a wider longitudinal region.) Although the high spectral width regions circled in these figures may very likely be a result of cusp precipitation, they do not necessarily highlight the full extent of the cusp. High spectral width values are observed within the polar cap at all magnetic local times (see the discussions and references in Chisham et al. (2008) [details above], and Chisham et al. (2007) – A decade of the Super Dual Auroral Radar Network (SuperDARN): scientific achievements, new techniques and future directions – Surv. Geophys., 28, 33-109 [specifically sect. 4, pages 60-67]). If Doppler spectral width is being used to estimate the location of the OCB then it is important to determine the spectral width boundary (SWB) location (see references in the same 2 papers). It is also important that spectral width values are only considered from radar beams that are aligned close to the meridional direction (see Chisham et al. (2005) – The accuracy of using the spectral width boundary measured in off-meridional SuperDARN HF radar beams as a

proxy for the open-closed field line boundary – Ann. Geophys., 23, 2599-2604).

(4) Quality and clarity of the figures containing the radar data:

The radar data plots in figures 2, 4, and 6 are incredibly messy, cluttered, and difficult to interpret, especially panels a and d, where line-of-sight (LOS) velocity and spectral width are displayed across the radar fields-of-view. These figures need to be simplified. Is all the LOS velocity data required in panel a? Are the merged vectors not information enough? Especially given that the LOS data on their own are open to severe misinterpretation. Can a boundary be determined from the spectral width data (see above) rather than highlighting a vague blob of high spectral width? If such a boundary was determined, then over-plotting this boundary on the velocity vector panels would be highly informative.

---

## Referee Comment (RC3) · Anonymous Referee #3 · 28 Aug 2018

Review of Zou et al for Annales Geophysicae, 28 Aug 2018

This manuscript uses a combination of satellite and ground-based radar data to estimate the spatial extent of magnetopause reconnection for 3 example events. The motivation for the study is very good and the results are potentially interesting and important but, in my view, the crucial radar analysis falls short of the state of the art and needs improving to support the interpretation. Even if this does not radically change the main results, it would put the results on a sounder footing, better evaluate sources and sizes of uncertainties, and allow the results given here to be compared more objectively to past and future studies. For this reason, I would not recommend publication in its present form. My recommendations are as follows:

1. Follow the state of the art

[Figure]

In the current analysis, evidence for the reconnection X-line is essentially based on looking for high-speed flows in the vicinity of a high radar spectral width region (e.g., Figure 2a-d) and the X-line extent is estimated from a longitudinal profile of northward velocity at a relatively arbitrary magnetic latitude. In my view this is a rather crude analysis and it should be possible to do this better by estimating the profile of the reconnection electric field itself along the open-closed field line boundary (OCB) and its time evolution following the methodology set out in detail in:

Chisham, G., et al. (2008), Remote sensing of the spatial and temporal structure of magnetopause and magnetotail reconnection from the ionosphere, Rev. Geophys., 46, RG1004, doi:10.1029/2007RG000223.

Freeman, M. P., G. Chisham, and I. J. Coleman (2007), Remote sensing of reconnection, in Reconnection of Magnetic Fields, edited by J. Birn and E. Priest, chap. 4.6, pp. 217–228, Cambridge Univ. Press, New York.

In essence, this method requires the following steps:

a. Identify the OCB objectively at as many locations as possible using available datasets and interpolate in space and time where necessary using suitable models, e.g., figures 6, 8, 9, 11 in Chisham et al (2008).

b. Estimate the reconnection electric field along the OCB by measuring the electric field component parallel to the boundary (or ExB velocity component perpendicular to it) in the rest frame of the generally moving boundary, e.g., figure 13 in Chisham et al. (2008).

c. Plot profiles of the reconnection electric field versus MLT over the time interval of interest. Use the zero crossing locations of these profiles to estimate the MLT extent of reconnection as a function of time, e.g., figure 7 of Pinnock et al., (2003), The location and rate of dayside reconnection during an interval of southward interplanetary magnetic field, Ann. Geophys., 21, 1467–1482.

d. Project the MLT extent to the magnetopause using a suitable model to estimate the X-line length and its evolution and to compare with in-situ spacecraft observations of presence or absence of reconnection, e.g., figure 8 of Pinnock et al. (2003).

The authors' analysis is only a very crude approximation to this. Particular areas of improvement that I would recommend include:

2. Improved estimates of the OCB (step 1a above)

a. The authors use a 150 m/s spectral width threshold to estimate the OCB but then apply it rather vaguely by drawing a red contour in figures 2d, 4d, 6d which doesn't match the 150 m/s threshold everywhere. The authors then largely ignore this anyway by using examining the ExB velocity on a fixed latitude circle that is generally poleward of where they say the OCB is. For example, for the first event in section 3.1.2, in lines 293-295 it is said that the OCB is at 77 deg latitude based on the spectral width in figure 2d but in lines 360-366 the 80 deg latitude circle is used as the OCB for the velocity cross-section shown in figure 2f. Similarly, in section 3.2.2, it is 77 deg latitude (lines 390-391) from figure 4d and 79 deg latitude (figure 4 caption) used for figure 4f. And in section 3.2.2, it is 80 deg latitude (figure 6 caption) used for figure 6g,h but the spectral width boundary is unstated and appears to be at lower latitude (at about the projected THA position).

b. According to the following references it should be possible to estimate the OCB from spectral widths at a wide range of local times using the method of Chisham and Freeman (2004) and I recommend that this be attempted more carefully and objectively.

Chisham, G., and M. P. Freeman (2003), A technique for accurately determining the cusp-region polar cap boundary using SuperDARN HF radar measurements, Ann. Geophys., 21, 983–996.

Chisham, G., and M. P. Freeman (2004), An investigation of latitudinal transitions in the SuperDARN Doppler spectral width parameter at different magnetic local times, Ann.

Geophys., 22, 1187–1202.

Chisham, G., M. P. Freeman, and T. Sotirelis (2004a), A statistical comparison of Su-perDARN spectral width boundaries and DMSP particle precipitation boundaries in the nightside ionosphere, Geophys. Res. Lett., 31, L02804, doi:10.1029/2003GL019074.

Chisham, G., M. P. Freeman, T. Sotirelis, R. A. Greenwald, M. Lester, and J.-P. Villain (2005a), A statistical comparison of SuperDARN spectral width boundaries and DMSP particle precipitation boundaries in the morning sector ionosphere, Ann. Geophys., 23, 733–743.

Chisham, G., M. P. Freeman, T. Sotirelis, and R. A. Greenwald (2005b), The accuracy of using the spectral width boundary measured in off-meridional SuperDARN HF radar beams as a proxy for the open-closed field line boundary, Ann. Geophys., 23, 2599–2604.

Chisham, G., M. P. Freeman, M. M. Lam, G. A. Abel, T. Sotirelis, R. A. Greenwald, and M. Lester (2005c), A statistical comparison of SuperDARN spectral width boundaries and DMSP particle precipitation boundaries in the afternoon sector ionosphere, Ann. Geophys., 23, 3645–3654.

c. The OCB can also be estimated from other data, such as DMSP particle pre-cipitation. It seems that this data might be available for the events studied, see https://heliophysicsdata.sci.gsfc.nasa.gov/websearch/dispatcher Even if not particu-larly close in MLT or UT it may be useful as a constraint.

d. The T89 model projections of the THA magnetopause crossing to the ionosphere in Figures 4 and 6 appear to agree with the OCB location estimated from the spectral width. It would thus seem reasonable to use the model to estimate the OCB location in the ionosphere at all dayside MLT at this UT.

The projected location of THE may be different in these two cases because from Figure 3 there is evidently a rapid outward expansion of the magnetopause from 9.4 RE to 10.2

RE between 1826 and 1828 UT which would need appropriate re-scaling of the model to capture, and in Figure 5 the spacecraft are separated by over 30 min in time and so again the model conditions are probably different. In these cases, and for the figure 2 event, it seems reasonable to explore simple scalings of the T89 model that would fit the magnetopause crossing location of each spacecraft and see if this improves the projected location of the spacecraft with respect to the spectral width boundary. If so, then the model could be used to extrapolate to all dayside MLT.

e. Alternatively, a simple offset circle model is commonly a good approximation to the OCB, whose free parameters could be constrained by spectral width and possibly DMSP data. This would at least be an improvement on assuming a latitudinal circle that is rather unrelated to the spectral width boundary.

In all of the above cases, limitations and assumptions can be assessed by error and sensitivity analyses. For example, how are the results 1b-d above affected by changing the inferred boundary by 1 degree say?

3. Take account of the generally moving OCB (step 1b above)

As emphasised in the references in 1 above, the reconnection rate is the electric field in the frame of the moving OCB and this can sometimes affect the inference of whether reconnection is occurring or not, e.g., see Figure 13 of Chisham et al. (2008). Some account of this should be taken in the present analysis as it may affect the edges of the inferred reconnection region in particular and hence the FWHM.

4. Project the ExB velocity perpendicular to the boundary (relevant to step 1c above)

Given the strong rotation of the flow seen in figure 2 in particular, consideration should be given of the effect of uncertainties in the assumed orientation of the OCB on the projected flow component across it as this could change the inferred X-line extent.

5. Improved consideration of the temporal evolution

The current analyses are strongly biased towards comparisons of magnetopause and

ionospheric observations of reconnection at a common instant. Given the uncertainties in how reconnection may evolve at the magnetopause, and the ionospheric response times, it would helpful to repeat the analysis shown in figure 2f, 4f, and 6g,h at some sampling frequency throughout the intervals shown in figures 2e, 4e, and 6e,f. The temporal evolution of los data shown in figures 2e, 4e, and 6e,f are a rather poor proxy by which to estimate the evolution of X-line extent and something similar to figure 7 of Pinnock et al (2003) would be very interesting to see, especially for the inferred complex evolution of the Apr 29 event.

6. Discrepancies in magnetopause to ionosphere projection (step 1d above)

The magnetopause crossings of spacecraft THA and THD in figure 2, and THE in figure 4 (and possibly figure 6 too) project several degrees of latitude away from the expected OCB location based on spectral width. This suggests that the estimation of X-line extent at the magnetopause from that inferred in the ionosphere will be in error because it is based on the same T89 model that seemingly incorrectly projects the satellite position to the ionosphere. As mentioned in 2d above, it would be helpful to try to estimate the uncertainty by considering whether there is some simple rescaling of the T89 model that would reduce the discrepancy in the magnetopause-to-ionosphere projection.

I would also add that the description of the mapping method given in lines 372-376 is too vague to allow others to reproduce your method. It also seems that you use the same T89 mapping factor of 55 for all three events, which seems questionable, e.g., solar wind dynamic pressure is 50% larger for Apr 19 event. It also implies that the factor is the same for all MLT which is unlikely I think, especially over the 10 Re magnetopause extent inferred for the Apr 29 event. Please could you improve your method description and assess the associated uncertainties.

7. I would recommend that you reference and discuss the following first 5 papers in lines 136-141 as these have done a similar comparison of simultaneous reconnection

evidence from space and ground to infer X-line length. I would also recommend that you consider the implications of these and the sixth reference to your discussion in section 3.4 as they seem to be relevant to the factors affecting X-line extent (e.g., IMF orientation, component or anti-parallel reconnection, turbulence):

Phan, T.D., Freeman, M.P., Kistler, L.M. et al. Earth Planet Sp (2001) 53: 619. https://doi.org/10.1186/BF03353281

Pinnock, M., G. Chisham, I. J. Coleman, M. P. Freeman, M. Hairston, and J.-P. Villain (2003), The location and rate of dayside reconnection during an interval of southward interplanetary magnetic field, Ann. Geophys., 21, 1467–1482.

Coleman, I. J., G. Chisham, M. Pinnock, and M. P. Freeman (2001), An ionospheric convection signature of antiparallel reconnection, J. Geophys. Res., 106, 28,995–29,007.

Chisham, G., I. J. Coleman, M. P. Freeman, M. Pinnock, and M. Lester (2002), Ionospheric signatures of split reconnection X-lines during conditions of IMF Bz < 0 and |By| $\sim$ |Bz|: Evidence for the antiparallel merging hypothesis, J. Geophys. Res., 107(A10), 1323, doi:10.1029/2001JA009124.

Chisham, G., M. P. Freeman, I. J. Coleman, M. Pinnock, M. R. Hairston, M. Lester, and G. Sofko (2004b), Measuring the dayside reconnection rate during an interval of due northward interplanetary magnetic field, Ann. Geophys., 22, 4243–4258

Coleman, I. J., and M. P. Freeman (2005), Fractal reconnection structures on the magnetopause, Geophys. Res. Lett., 32, L03115, doi:10.1029/2004GL021779.

---

## Author Comment (AC1) · 27 Sep 2018

Reviewer comment: The authors discuss conjugate observations of two THEMIS satellites crossing the magnetopause in short succession, with ground based radar observations to determine the lengths of a dayside reconnection line at the magnetopause. The methodology seems to be interesting and the paper is well written with a very good introduction to the general problem. However I have some issues with the current data analysis and the event selection that are significant enough to not recommend publication at this time. General Point: As the authors admit, their determined length of the X-line will be limited to the longitudinal coverage of the radars. This unavoidable limitation will always significantly influence their conclusion about the length of the "actual" X-line, which could be considerably longer, and will prevent them from ever finding

a global answer. That will seriously limit the usefulness of the methodology, thought generally its an interesting approach.

Response: We realize that the term "X-line extent" in our manuscript has caused confusion. The X-line the reviewer refers to is the magnetic geometry along which reconnection occurs at various rates and frequencies, which is indeed considerably longer than the radar coverage. However, our study intends to focus on the extent of reconnection bursts. We have revised the term "X-line extent" to "reconnection burst extent" as we do not aim at determining the extent of the global X-line but the localized bursty reconnection in the area of satellite-ground conjunction. Other bursts could occur outside the radar and satellite coverage but those are beyond the focus of this research. Please see also the response to the next comment. With this clarification, the radar longitudinal coverage is sufficiently large for the purpose of this study. For example, the ionospheric flow structures under examination have a skewed Gaussian-shape velocity profile (Figures 2e, 4e, and 6e), and the FWHM of the profile is located completely within the radar coverage.

Reviewer comment: About the introduction: There are several significant publications using IMAGE/FUV observations. This mission had the ability to observe emissions from precipitating (cusp) ions over the entire polar region at once and was therefore not limited like the radar coverage in the present manuscript. Studies using these data have shown evidence that during southward IMF conditions the entire dayside is open leading to very long dayside reconnection lines. So, based on these results the length of the X-line is not the driving question. In additions, decades of cusp observations in all local time sectors show precipitating ions. X-lines in general seem to be very long. Cusp observations have shown that a substantial part of reconnection is dominated by pulsed reconnection [Lockwood et al.,. . ...]. The question is therefore – is the long X-line pulsing as "One" or are individual longitudinal sections have their own pulsation frequency? That should lead to scenarios presented in this manuscript, sections of X-lines that are active next to sections of X-lines temporarily inactive. This is how I

would interpret the observations in the manuscript. Therefore the conclusion would not be about the length of the X-line since that would be masked by the temporal nature of the reconnection process, which might lead to misleading results. In any case, I was surprised that there was no reference to this rather ground breaking IMAGE observations anywhere. These observations [e.g., Fuselier et al., 2002] should be added in the introduction and properly described.

Response: As the reviewer inferred we examine bursts of reconnection. Our study shows that a reconnection burst is not necessarily a pulse of a long X-line but can occur over a finite area. IMAGE observations have provided global configuration of reconnection where reconnection bursts are embedded. The global-scale reconnection configuration is not the focus of this study but it offers valuable groundwork of clarifying the scope of the research. We rewrite the first paragraph as "...Reconnection tends to occur at sites of strictly anti-parallel magnetic fields as anti-parallel reconnection [e.g. Crooker, 1979; Luhmann et al., 1984], or occur along a line passing through the sub-solar region as component reconnection [e.g. Sonnerup, 1974; Gonzalez and Mozer, 1974]. Evidence shows either or both can occur at the magnetopause and the overall reconnection extent can span from a few to 40 Re [Paschmann et al., 1986; Gosling et al., 1990; Phan and Paschmann, 1996; Coleman et al., 2001; Phan et al., 2001, 2003; Chisham et al., 2002, 2004, 2008; Petrinec and Fuselier, 2003; Fuselier et al., 2002, 2003, 2005, 2010; Petrinec and Fuselier, 2003; Pinnock et al., 2003; Bobra et al., 2004; Trattner et al., 2004, 2007, 2008, 2017; Trenchi et al., 2008]. However, reconnection does not occur uniformly across this configuration but has spatial variations [Pinnock et al., 2003; Chisham et al., 2008]. The local time extent of reconnection bursts is the focus of this study."

Reviewer comment: Specific Points: Line 188: the D-shaped distribution do not persist into the ionosphere due to the conservation of the first adiabatic invariant. The D shape changes into a Crescent shape as soon as the ambient B field increases, which it definitely will in the cusps. This has been observed in the cusp regions for decades.

This effect is so pronounced that it can be even used directly at the magnetopause. The "bending over" of the D-shape distribution observed during magnetopause crossings has been used in a recent study by Broll et al. (2017) (JGR) to determine the distance to the X-line from the MMS satellites and infer the X-line location. Cusp Steps have nothing to do with D-shape distributions. Cusp steps are the result of changes in the reconnection rate at the magnetopause or caused by spatially separated X-lines. Cusp-steps have been discussed in great detail by Lockwood and Smith in the 90ties as manifestation of pulsed reconnection leading to the pulsed reconnection model and by e.g., Onsager et al [1995] or Trattner et al. [2002] as spatially separated X-lines.

Response: We agree with the reviewer and correct the statement as "The D-shaped ion distributions are deformed into a crescent shape as ions travel away from the reconnection site [Broll et al., 2017]". We also replace case study #1 with a new event and the new event has a distorted D-shaped distribution. Details can be found below.

Reviewer comment: The authors use patchy reconnection also in the case of spatially separated X-line or partial X-lines. This will be a source of confusion for colleagues not too familiar with the subject. Patchy reconnection usually describes pulsed reconnection – temporal changes in reconnection. While the authors do a reasonable good job in trying to keep the temporal and spatial regimes apart I would recommend to revisit that issue throughout the paper.

Response: We follow the reviewer's suggestion and add "the term patchy has also been used to describe the temporal characteristics of reconnection [e.g. Newell and Meng, 1991]. But this paper primarily focuses on the spatial properties". We use "spatially patchy reconnection" to replace "patchy reconnection" throughout the text.

Reviewer comment: Figure 2: The symbol for Th-D is completely invisible – if it wasn't for Figure 4 I would not have realized that there are indeed two separate magnetic foot points in that plot. Chose a different more prominent color.

Response: As advised by the reviewer we replace this event with an event that has

good field line mapping. Please find the attachment for the new event.

Reviewer comment: Figure 2: it is mentioned in line 209 – the satellite foot points should map close to the radars FOV. I would recommend that the authors look for events where the satellite foot points are actually in the FOV of the radars to make absolutely sure that these observations are linked. Throughout the paper but especially in Figure 2 I do not have the impression that this is the case which makes the data analysis rather questionable. Therefore I fail to see how the observed D-shape distributions at the magnetopause are connected with particular flow channels which is the essential part of the study. The authors also mark the cusp foot point in the radar images. Discussing again the events in figure 2, Th-D clearly saw an ion jet. It therefore observed reconnection at the magnetopause and was on a newly opened field line. The D shape distribution, while looking a bid crooked compared to the other D-shape distributions in the manuscript, travels along the magnetic field. The magnetic field, at that time the distribution was observed, was still northward. Therefore the satellite was in the LLBL and the ions move toward the northern cusp where the radar observations observe flow channels. All open magnetopause field lines map into the cusps. So the Th-D magnetic foot point, were the D-distribution was observed, should be in that region marked as cusp in figure 2d. It is not, its not even in the FOV for the radar.

Response: To address reviewer's comment, we replace Figures 1-2 (see Figures 1-2). In the new event the footprint is within the radar FOV and close to the open-closed field line boundary. The corresponding text is changed to the following.

[revised manuscript text omitted]
 the presence of two dark red (>220 m/s spectral width) regions embedded within the $\sim$200-m/s spectral widths (circled in red, the red dashed line is due to the discontinuous backscatters outside the INV FOV), corresponding to the two flows.

Figures 2a-c all observed a channel of fast anti-sunward flow in the pre-noon sector of the high latitude ionosphere, and the flow had a limited azimuthal extent. If the flow corresponded to magnetopause reconnection, the X-line is expected to span over a limited local time range. This is consistent with the THEMIS satellite observation in Section 3.1.1, where THE at Y = -2.9 Re detected clear reconnection signatures, while THA at Y = -4.8 Re did not. In fact, if we project the satellite location to the

ionosphere through field line tracing under the T89 model, THE was positioned at the flow longitude, while THA outside the flow was to the west (Figure 2a).

While this paper primarily focuses on the spatial extent of reconnection bursts, the temporal evolution of reconnection can be obtained from the time series plot in Figure 2e. Figure 2e presents the INV LOS measurements along 80° MLAT (just poleward of the open-closed field line boundary with good LOS measurements) as functions of magnetic longitude (MLON) and time. Similar to the snapshots, the color represents LOS speeds that project to the anti-sunward direction, and the flow of our interest appears as a region of red color. The time and the location where THA and THE crossed the magnetopause are marked by the vertical and horizontal lines. The flow emerged from a weak background at 2120 UT and persisted for ∼30 min in INV FOV. At the onset the flow eastern boundary was located at -82° MLON, and interestingly, this boundary spread eastward with time in a similar manner as events studied by Zou et al. [2018]. The flow western boundary was located around -77° MLON during 2120-2134 UT, and started to spread eastward after 2134 UT. Hence the reconnection-related ionospheric flow, once formed, has spread in width and displaced eastward. The spreading has also been noticed in the other two events (see Section 3.3), indicating that this could be a common development feature of the reconnection-related flows. The spreading was fast in the first 6 min and then slowed down stabilizing at a finite flow extent (until the eastern boundary went outside FOV at 2134 UT).

A consequence of the flow temporal evolution is that THA, which was previously outside the reconnection-related flow, became immersed in the flow from 2130 UT, while THE, which was previously inside the flow, was left outside from 2142 UT (Figure 2e). This implies that at the magnetopause the reconnection has spread azimuthally sweeping across THA, and has slid in the –y direction away from THE. This is in perfect agreement with satellite measurements shown in Figures 2q-z. Figures 2q-z presents subsequent magnetopause crossings made by THA and THE following the crossings in Figures 2g-p. THA detected an Alfvenic reconnection jet and a clear D-shape ion

distribution, and THE detected a jet much slower than the Alfvenic speed and an ion distribution without a clear D-shape. This corroborates the connection between the in-situ reconnection signatures with the fast anti-sunward ionospheric flow, and reveals the dynamic evolution of reconnection in the local time direction.

We quantitatively determine the flow extent in Figure 2f. Figure 2f shows the INV LOS velocity profile at 2125 UT as a function of magnetic longitude and distance from 0° MLON. The 2125 UT is the same time instance as in Figures 2a-c and is the time when the flow extent has slowed down from spreading and stabilized. The profile is taken along 80° MLAT. While this latitude is 2° poleward of the open-closed field line boundary, the shape of the flow did not change much over the 2° displacement and thus still presents the reconnection extent. The flow velocity profile has a skewed Gaussian shape, and we quantify the flow azimuthal extent as the full-width-at-half-maximum (FWHM). The FWHM was 13° in MLON or 260 km at an altitude of 260 km. Also shown is the SECS velocity profile. Here we only show the northward component of the SECS velocity as this component represents reconnecting flows across an azimuthally-aligned open-closed field line boundary. The SECS velocity profile gives a FWHM of 13.5° in MLON or 270 km, very similar to the LOS profile.

It is noteworthy mentioning that the velocity profile obtained above approximates to the profile of reconnection electric field along the open-closed field line boundary (details in Figure S3). Reconnection electric field can be estimated by measuring the flow across the open-closed field line boundary in the reference frame of the boundary [Pinnock et al., 2003; Freeman et al., 2007; Chisham et al., 2008]. However, a precise determination of the boundary motion is subject to radar spatial and temporal resolution and for a slow motion like events studied in this paper (Figure S1), the signal to noise ratio is lower than one. For this reason this paper focuses on the velocity profile poleward of the open-closed field line boundary, which is less affected by the error associated with the boundary.

To infer the reconnection extent at the magnetopause, we project the flow width in the

ionosphere to the equatorial plane. The result suggests that the reconnection local time extent was ∼3 Re.

Before closing this section, we would like to point out that the determined extent is characterized by the FWHM of the fast anti-sunward ionospheric flow, which allows weak flows to extend beyond the flow extent. When THA and THE were positioned within the weak flows in the ionosphere, they at the magnetopause observed flows much weaker than the Walen prediction. This may imply that there were two components of reconnection at different scales in this event: weak background reconnection signified by the slow flows, and embedded strong reconnection bursts signified by the fast flows.

Reviewer comment: Line 338: One of the open questions in magnetic reconnection is still how the reconnection rate develops along the length of the X-lines. Since decades of research showed that pulsed reconnection is a rather significant process, it is conceivable that individual sections along a "long" X-line pulse at different frequencies. I therefore would expect that it is very likely that magnetopause crossings by multiple satellites show active and temporarily inactive sections along an X-line. This is not prove that a dayside X-line is short. The interpretation of the authors that this event is a spatially restricted X-line based on flow channels at very different latitudes is not convincing, especially since the satellite observations are outside the flow channels for which observations exist. I also want to stress that in the pulsed reconnection model, field lines that were opened before reconnection briefly stopped, are convecting and provide a continuous transfer of magnetosheath plasma into the magnetosphere. That should certainly influence your radar observations. It is unlikely that the ionosphere would respond that quickly to short changes in the reconnection rate. The magnetosphere is generally rather slow in its response to outside changes. That will make linking ionospheric flow channels to magnetopause observations rather challenging. Radar observations of ionospheric convection, direction and velocities, are often used to estimate global convection pattern in the polar ionosphere using various models. These "convection cells" could be overlaid in the radar plots to make a connection

between the satellite magnetic foot points outside the radar FOV and the radar data. Depending on how these global convection cells look like they might provide a more convincing picture that these observations are actually linked.

Response: We have replaced Figures 1-2 to the new event where the satellite footprints were within the radar FOV and close to the open-closed field line boundary. We believe that this event provides a more convincing case for establishing the space and ground connection.

We agree with the reviewer that reconnection can happen over various temporal scales but the typical time scale of reconnection bursts, or FTEs is found to be a few minutes [Lockwood and Wild, 1993; Kuo et al., 1995; Fasel, 1995]. This can be resolved by radars considering that M-I coupling time scale on the dayside is ∼1-2 min [e.g. Carlson et al., 2004]. Studies have compared the time scale of ionospheric flows with FTEs and found a very similar distribution [McWilliams et al., 1999], suggesting that ionospheric flows well capture reconnection variability at least down to FTE time scale. We add the following text to the end of the methodology section.

"Note that reconnection can happen over various spatial and temporal scales and our space-ground approach can resolve reconnection bursts that are larger than 0.5 Re and persist longer than a few minutes. This is limited by the radar spatial and temporal resolution, and the magnetosphere-ionosphere coupling time which is usually 1-2 min [e.g. Carlson et al., 2004]. This constraint is not expected to impair the result because reconnection bursts above this scale have been found to occur commonly in statistics (see the Introduction section for spatial and Lockwood and Wild [1993], Kuo et al. [1995], Fasel [1995], and McWilliams et al. [1999] for temporal characteristics)."

We have followed the reviewer's opinions and added the global convection pattern in supporting Figure S2. The radars employed in the paper has contributed to the majority of the backscatter on the dayside and including more radars do not change the conclusion. Again we focus on the extent of individual reconnection-related flow,

not the sum of all the flows on the dayside. It may also noteworthy to point out an important difference between our study and previous studies: our events occurred under non-storm time, where the open-closed field line is confined within the utilized few radar FOVs, while previous studies using a wider network of SuperDARN radars focus on storm time period where the boundary has expanded to low latitude.

**2013 Feb 2**

(a)

IMF (nT)

|B|  Bx  By  Bz

UT    2120    2130    2140    2150

(b)

Y_gsm (Re)

THE
THA

X_gsm (Re)

**Fig. 1.** Figure 1a: OMNI IMF condition on Feb 2, 2013. Figure 1b: THE and THA locations projected to the GSM X-Y plane. The inner curve marks the magnetopause and the outer curve marks the bow shock.

[Figure]

**Fig. 2.** Figure 2a: SuperDARN LOS speeds (color tiles) and merged velocity vectors (color arrows) in the Altitude adjusted corrected geomagnetic (AACGM) coordinates. The FOVs of the RKN, INV, and CLY radars ar

**2015 Apr 19**

**(a)**

IMF (nT)

|B| Bx By Bz

UT    1815    1825    1835

**(b)**

Y_gsm (Re)

THA

THE

X_gsm (Re)

**Fig. 3.** Figure 3. OMNI IMF condition and THEMIS satellite locations on Apr 19, 2015 in a similar format to Figure 1.

**Fig. 4.** Figure 4. THEMIS and SuperDARN measurements of reconnection bursts on Apr 19, 2015 in a similar format to Figure 2. The velocity time evolution in Figure 4e and the velocity profile in Figure 4f are t

**Fig. 5.** Figure 5. OMNI IMF condition and THEMIS satellite locations on Apr 29, 2015 in a similar format to Figure 1.

[Figure]

**Fig. 6.** Figures 6a-d: SuperDARN measurements of reconnection bursts on Apr 29, 2015 in a similar format to Figures 2a-d except that in Figure 6a the color of the CLY color tiles represent LOS speeds towards t

[Figure]

**Fig. 7.** Figure S1. Location of the open-closed field line boundary (marked by the black dashed line) in the three studied events. The open-closed field line boundary is determined based on the spectral width

**Fig. 8.** Figure S2. Global convection maps of the three studied events. The SHF velocities are shown as color arrows, and the contours of the electric potential are shown as black solid (at the duskside) and d

**Fig. 9.** Figure S3. Reconnection electric along the open-closed field line boundary for the Feb 02, 2013 event. Figures S3a-c: snapshots of spectral width measurements around the space-ground conjunction time

---

## Author Comment (AC2) · 27 Sep 2018

This paper is concerned with estimating the extent of reconnection X-lines on the Earth's magnetopause, with an overall aim of measuring, and understanding spatial and temporal variability in magnetic reconnection. For studies of this type, conjugate observations combining spacecraft and ground-based measurements can be very important. There are some aspects of reconnection (such as the localised plasma physics) that can only be measured by in-situ spacecraft. There are also some aspects (such as the macrophysics of the process) that can only be measured by instruments that provide a wider view, such as auroral imagers or ground-based radars. However, the local time extent of reconnection regions can only be determined unambiguously using ionospheric measurements (in the absence of a massive armada of spacecraft).

[Figure]

Similarly, the amount of flux transfer occurring during reconnection can only be determined unambiguously using ionospheric measurements. And consequently, the patchy (spatial variation) and bursty (temporal variation) of reconnection can only be unambiguously studied using ionospheric measurements. To measure the extent of reconnection from ionospheric measurements (which can then be mapped back to the magnetopause) first requires the identification of the ionospheric footprint of the open-closed magnetic field line boundary (OCB). The regions where the ionospheric plasma flow crosses this boundary (in the frame of the boundary – which is typically in motion itself) map to the regions on the magnetopause where reconnection is occurring. Although the text shows that the authors appear to appreciate this, they do not analyse their ionospheric data in this way. Consequently, I have some major issues with the introductory text and the radar data analysis and presentation. The authors need to address these major points before the paper can be reviewed properly. (1) Some of the background referencing is misdirected and inadequate: The referencing of spacecraft observations associated with reconnection (extending from lines 95 to 117) starts with the phrase – 'The extent of reconnection X-lines has been observationally determined based on fortuitous satellite conjunctions. . .'. This is not true. Even if the word 'determined' was changed to 'estimated' it would still be a stretch of the truth. The 'extent of reconnection X-lines' cannot be unambiguously determined (or even estimated) from spacecraft observations. Interpretations of multiple spacecraft observations still have to make the assumption that the X-line is continuous between spacecraft, or that it is not continuous between spacecraft. X-lines may also continue longitudinally outside of the view of the spacecraft. All that multiple spacecraft measurements can do (given that the assumptions made are correct) is provide upper or lower limits on the X-line extent.

Response: We completely agree with the reviewer's opinion on the limitations of spacecraft observations. Those limitations are the exact motivation of adopting the space-ground approach in this paper as mentioned in the introduction section. We change the statement to "studies have attempted to constrain the extent of reconnection X-lines

based on fortuitous satellite conjunctions". The word "constrain" has been used by the paper "Spacecraft measurements constraining the spatial extent of a magnetopause reconnection X line" by Walsh et al. 2017.

The referencing of ionospheric observations associated with reconnection (extending from lines 118 to 141) concentrates on those related mainly to local (often single radar) measurements of fast anti-sunward flows observed by radar (such as pulsed ionospheric flows [PIFs]) and their auroral counterpart (poleward-moving auroral forms [PMAFs]). These typically occur within the polar cap, and not necessarily at the ionospheric footprint of the OCB. Although all these observations are of phenomena that are consequences of reconnection, and which provide important information about the patchy and bursty nature of reconnection (and links to FTEs, etc.), they don't allow the unambiguous estimation of the extent of the X-line. Hence, many of these references are actually superfluous to the paper. As mentioned above, to measure the extent of the reconnection X-line in the ionosphere requires the identification of the footprint of the OCB and the region for which there is plasma flow across it. (Although, similar caveats to the spacecraft observations also exist if there is not complete longitudinal coverage covering the whole ionospheric projection of the X-line.) There are a large number of papers that have studied and measured reconnection in this way that are not mentioned in the introduction of the present paper. A significant reference that reviews most of the work in this area, as well as outlining the techniques required to make these measurements, is Chisham et al. (2008) – Remote sensing of the spatial and temporal structure of magnetopause and magnetotail reconnection from the ionosphere – Rev. Geophys., 46, RG1004. Other papers that have measured the extent of the reconnection X-line using these methods include; (i) Pinnock et al. (2003) – The location and rate of dayside reconnection during an interval of southward interplanetary magnetic field – Ann. Geophys., 21, 1467-1482, which studied the same event that was observed in Equator-S data by Phan et al. (2000). They estimated the length of the reconnection X-line on the dayside magnetopause at this time to be âĹij38 Re based on the 10 hours of local time that flow was observed crossing the OCB in the

ionosphere. (ii) Chisham et al. (2004) – Measuring the dayside reconnection rate during an interval of due northward interplanetary magnetic field – Ann. Geophys., 22, 4243-4258, which measured the X-line extent of lobe reconnection during northward IMF to be âĹij6-11 Re.

Response: We thank the reviewer for the important references. We realize that the term "X-line extent" in our manuscript has caused confusion. In our original terminology we used "magnetic separator" to refer to the global configuration along which reconnection occurs at various rates, and used "X-lines" to refer to regions of strong reconnection, i.e., reconnection bursts. Such usage has been common in the literature (especially in FTE studies [e.g., Fear et al., 2008, 2010] and local numerical simulations [e.g., Shay et al., 2003; Sheperd and Cassak, 2012]). But to avoid confusion we replace "extent of X-lines" with "extent of reconnection bursts" throughout the text. Therefore the title of the paper is "local time extent of magnetopause reconnection bursts using space-ground coordination". Similar changes are made throughout the text.

The references suggested by the reviewer provide valuable groundwork of clarifying the scope of this study. We rewrite the first paragraph as "...Reconnection tends to occur at sites of strictly anti-parallel magnetic fields as anti-parallel reconnection [e.g. Crooker, 1979; Luhmann et al., 1984], or occur along a line passing through the subsolar region as component reconnection [e.g. Sonnerup, 1974; Gonzalez and Mozer, 1974]. Evidence shows either or both can occur at the magnetopause and the overall reconnection extent can span from a few up to 40 Re [Paschmann et al., 1986; Gosling et al., 1990; Phan and Paschmann, 1996; Coleman et al., 2001; Phan et al., 2001, 2003; Chisham et al., 2002, 2004, 2008; Petrinec and Fuselier, 2003; Fuselier et al., 2002, 2003, 2005, 2010; Petrinec and Fuselier, 2003; Pinnock et al., 2003; Bobra et al., 2004; Trattner et al., 2004, 2007, 2008, 2017; Trenchi et al., 2008]. However, reconnection does not necessarily occur uniformly across this configuration but has spatial variations [Pinnock et al., 2003; Chisham et al., 2008]. The local time extent of

Interactive
comment

reconnection bursts is the focus of this study."

(2) Identification of the extent of the reconnection region from fast ionospheric flows is flawed: Lines 52-54 state – 'The extent has also been inferred by radars as fast ionospheric flows moving anti-sunward across the open-closed field line boundary, but whether a particular ionospheric flow results from reconnection needs to be confirmed.' Firstly, the measured flows do not need to be fast. The fast flows highlighted in the paper are obviously driven by reconnection but these are predominantly polar cap flows (relating to the newly-opened flux tubes moving over the polar regions towards the nightside), not flows at and across the OCB. Any flow across the OCB, whether fast or slow, implies that reconnection has occurred, as closed flux has been converted to open flux. By the same argument, if flow across the OCB is measured, spacecraft measurements are not required to prove that this flow is a result of reconnection (hence I disagree with the statement on lines 132-135). Lines 198-206 detail the SuperDARN radars used in the study. What I do not understand is why the authors restricted their study to only a few of the northern hemisphere radars when there is a much wider network of northern hemisphere SuperDARN radars that would provide a much greater longitudinal coverage? Larger coverage provides a much better global picture of the ionospheric convection and hence the reconnection driven flows across the OCB.

Response: We agree that conceptually ionospheric flows moving across the OCB, even slow, should be related to reconnection. However, to the best of our knowledge, there has been no confirmation of whether weak ionosphere flows meet the quantitative in-situ magnetopause reconnection criteria and our event #1 (updated as seen in the attachment) suggests that they actually correspond to plasma jets at the magnetopause much slower than the Alfven speed. Thus the slow ionospheric flows do not meet the in-situ definition of reconnection but should be treated separately. The focus of this paper is on strong bursts of reconnection. But our study, as well as Chisham et al. [2008], may have suggested that there are two components of reconnection at different scales: weak background reconnection signified by the slow flows, and em-

bedded strong reconnection bursts signified by the fast flows.

To avoid confusion, we replace the sentence as "The validity of the assumption can be tested by radars via examining ionospheric flows moving anti-sunward across the open-closed field line boundary".

The PolarDARN radars utilized in the paper have provided sufficient coverage for studying reconnection bursts in the area of satellite-ground conjunction. Reconnection bursts may also activate outside the radar FOV, but those are not the focus of the satellite-ground conjunction study and the terminology change mentioned above clarifies that this paper is not meant to determine the global X-line extent but individual reconnection burst extent. Backscatters from radars at lower latitudes were limited (see Figure S2) because the cusp, and the associated ionospheric irregularities, occurred at relatively high latitude ($>77-78°$ MLAT). It is noteworthy to point out the studied events occurred under non-storm time, while previous studies using a wide network of Super-DARN radars focus on storm time period where the OCB has expanded to low latitude.

Lines 297-298 state – 'The extent is determined at half of the maximum flow speed, which was âĹij400 m/s'. Why? There is still flow across the boundary outside this region that results from reconnection. Consequently, the dashed magenta lines in figures 2, 4, and 6 mean nothing, except to nicely frame the fast poleward flows into the polar cap. In a similar vein, lines 366-367 state 'We quantify the flow azimuthal extent as the full-width-at-half-maximum (FWHM) of the velocity profile'. Why? Any poleward flow (across the OCB) represents the creation of newly reconnected flux. In all 3 examples there are significant poleward flows east of the dashed magenta lines. In figures 2e and 2f the flow extent is 'quantitatively determined' using measurements at 80 degrees latitude. Why use the flows at this latitude to determine the longitudinal extent when they are well within the polar cap? These are not the same as the flows at the OCB latitude, and hence they do not show the longitudinal extent of reconnection. Hence, they cannot be reliably used to estimate the length of the X-line.

[Figure]

Response: We appreciate the reviewer's comment. As clarified above, we focus on reconnection bursts, which appear as fast anti-sunward flows in the ionosphere. It has been a common approach to measure the reconnection burst extent as the flow extent at a latitude poleward of the OCB [Goertz et al., 1985; Pinnock et al., 1993, 1995; Provan and Yeoman, 1999; Thorolfsson et al., 2000; McWilliams et al., 2001a, 2001b; Elphic et al., 1990; Denig et al., 1993; Neudegg et al., 1999, 2000; Lockwood et al., 2001; Wild et al., 2001, 2003, 2007; McWilliams et al., 2004; Zhang et al., 2008]. Slow flows have been allowed to extend beyond the boundaries of the fast flows [Mcwilliams et al., 2004], and we have clarified how fast and slow ionosphere flows are contrasted in terms of in-situ flows above. Since the longitudinal profile of the flow velocity has a skewed Gaussian shape, we have used FWHM. The use of FWHM is analogous to the methodology of Shay et al. [2003], who define reconnection as regions where the current density is larger than half of what is carried by the electron Alfven speed. This is clarified in the text.

We have compared our flow velocity profile with the reconnection electric field at the OCB in Figure S3. Figures S3a-c present the OCB (dashed black line) of the first case study around the space-ground conjunction time and longitude following Chisham and Freeman [2003, 2004] and Chisham et al. [2004, 2005a, 2005b, 2005c]. The OCB was nearly along a constant latitude. Figures S3d-f present time series of the spectral width measurements along beams 4, 7, and 10, as a function of latitude. The time series plot allows us to determine the speed of the OCB motion and we determined the speed at each individual beam. Figure S3g presents the electric field along the OCB in the frame of the ionosphere (dotted), and in the frame of the OCB (solid). The latter is the reconnection electric field. The reconnection electric field had essentially the same FWHM as the flow slightly poleward of the OCB (difference being less than the radar spatial resolution).

We note that the process of tracking OCB motion can introduce large uncertainties, especially for our events where the OCB moved very slowly (Figure S1). Given the radar

spatial ($\sim$0.3°) and temporal (2 min) resolution, the speed of OCB has an uncertainty of $\sim$300 m/s. This results in a signal to noise ratio generally around or even below one, even though we have not yet considered the measurement error associated with spectral widths or the error of using 150 m/s as the OCB threshold in any given event. A similarly poor signal to noise ratio has been found in Chisham et al. [2008]. This would affect the estimate of the electric field and would reduce the confidence of the results. The flow velocity poleward of the OCB is less affected by the OCB uncertainties.

Given that the electric field profiles at the OCB latitude and the flow velocity profile slightly poleward are about the same, that the echoes are more continuous at higher latitudes, and that our approach is consistent with a number of past works cited above, we think that our approach is sufficient to lead to the conclusion.

(3) The open-closed field line boundary (OCB) in the ionosphere is insufficiently determined: Lines 390-391 state 'The flow crossed the open-closed field line boundary at 77 degrees MLT. . .'. The determination of the OCB location is not clearly outlined anywhere or displayed clearly on the figures. Indeed, the OCB location in figures 2, 4, and 6 is never sufficiently determined (or visually presented) so it is impossible to know what the longitudinal extent of flows across the boundary is. The boundary is vaguely discussed as being the equatorward edge of the cusp, which is identified in these figures as being co-located with regions of high Doppler spectral width. (In actuality, comparing figures 2c and 2d, the poleward flow at the equatorward edge of the cusp is slower than that within the polar cap, and most likely extends over a wider longitudinal region.) Although the high spectral width regions circled in these figures may very likely be a result of cusp precipitation, they do not necessarily highlight the full extent of the cusp. High spectral width values are observed within the polar cap at all magnetic local times (see the discussions and references in Chisham et al. (2008) [details above], and Chisham et al. (2007) – A decade of the Super Dual Auroral Radar Network (SuperDARN): scientific achievements, new techniques and future directions – Surv. Geophys., 28, 33-109 [specifically sect. 4, pages 60-67]). If Doppler spectral

width is being used to estimate the location of the OCB then it is important to determine the spectral width boundary (SWB) location (see references in the same 2 papers). It is also important that spectral width values are only considered from radar beams that are aligned close to the meridional direction (see Chisham et al. (2005) – The accuracy of using the spectral width boundary measured in off-meridional SuperDARN HF radar beams as a C5 ANGEOD Interactive comment Printer-friendly version Discussion paper proxy for the open-closed field line boundary – Ann. Geophys., 23, 2599-2604).

Response: The references provided by the reviewer are highly relevant and have been included in the text. The OCB is determined as the 150 m/s spectral width boundary [e.g., Baker et al., 1995, 1997; Chisham and Freeman, 2003] as indicated in the text although we did not present the details in our previous manuscript. The details are now displayed in Figures S1 and S3. We also mark this boundary in Figures 2, 4, and 6 as a black dashed line.

We agree with the reviewer that high spectral width can span across a wide range. But here we look for structures embedded in the spectral width because the existence of a localized enhancement indicates enhanced energy input from the magnetosphere over a finite area. This is consistent with our focus on reconnection bursts. The "cusp" feature we refer to follows the dynamic cusp model where the cusp precipitation is driven by reconnection bursts. To avoid confusion with the traditional cusp, we rename it as enhanced soft electron precipitation.

(4) Quality and clarity of the figures containing the radar data: The radar data plots in figures 2, 4, and 6 are incredibly messy, cluttered, and difficult to interpret, especially panels a and d, where line-of-sight (LOS) velocity and spectral width are displayed across the radar fields-of-view. These figures need to be simplified. Is all the LOS velocity data required in panel a? Are the merged vectors not information enough? Especially given that the LOS data on their own are open to severe misinterpretation. Can a boundary be determined from the spectral width data (see above) rather than highlighting a vague blob of high spectral width? If such a boundary was determined, then

over-plotting this boundary on the velocity vector panels would be highly informative.

Response: We thank the reviewer for the suggestion and have simplified panels a and d by deleting isolated LOS backscatters and minimizing the overlap of backscatters. The OCB has also been overlaid on panels a, b, and c. We mentioned about determination of spectral width boundary. The red blobs in Figures 2d, 4d, and 6d highlight structures of high spectral width which are not related to the OCB determination but enhanced soft electron precipitation.

**2013 Feb 2**

(a)

IMF (nT)

|B| Bx By Bz

UT  2120  2130  2140  2150

(b)

Y_gsm (Re)

THE
THA

X_gsm (Re)

**Fig. 1.** Figure 1a: OMNI IMF condition on Feb 2, 2013. Figure 1b: THE and THA locations projected to the GSM X-Y plane. The inner curve marks the magnetopause and the outer curve marks the bow shock.

**Fig. 2.** Figure 2a: SuperDARN LOS speeds (color tiles) and merged velocity vectors (color arrows) in the Altitude adjusted corrected geomagnetic (AACGM) coordinates. The FOVs of the RKN, INV, and CLY radars ar

**2015 Apr 19**

**(a)**

IMF (nT)

|B| Bx By Bz

UT    1815    1825    1835

**(b)**

Y_gsm (Re)

THA

THE

X_gsm (Re)

**Fig. 3.** Figure 3: OMNI IMF condition and THEMIS satellite locations on Apr 19, 2015 in a similar format to Figure 1.

**Fig. 4.** Figure 4. THEMIS and SuperDARN measurements of reconnection bursts on Apr 19, 2015 in a similar format to Figure 2. The velocity time evolution in Figure 4e and the velocity profile in Figure 4f are t

**2015 Apr 29**

(a)

IMF (nT)

|B|  Bx  By  Bz

UT  1820  1840  1900  1920  1940

(b)

Y_gsm (Re)

THA  THE

X_gsm (Re)

**Fig. 5.** Figure 5. OMNI IMF condition and THEMIS satellite locations on Apr 29, 2015 in a similar format to Figure 1.

[Figure]

**Fig. 6.** Figures 6a-d: SuperDARN measurements of reconnection bursts on Apr 29, 2015 in a similar format to Figures 2a-d except that in Figure 6a the color of the CLY color tiles represent LOS speeds towards t

**Fig. 7.** Figure S1. Location of the open-closed field line boundary (marked by the black dashed line) in the three studied events. The open-closed field line boundary is determined based on the spectral width

**Fig. 8.** Figure S2. Global convection maps of the three studied events. The SHF velocities are shown as color arrows, and the contours of the electric potential are shown as black solid (at the duskside) and d

**Fig. 9.** Figure S3. Reconnection electric along the open-closed field line boundary for the Feb 02, 2013 event. Figures S3a-c: snapshots of spectral width measurements around the space-ground conjunction time

---

## Author Comment (AC3) · 27 Sep 2018

This manuscript uses a combination of satellite and ground-based radar data to estimate the spatial extent of magnetopause reconnection for 3 example events. The motivation for the study is very good and the results are potentially interesting and important but, in my view, the crucial radar analysis falls short of the state of the art and needs improving to support the interpretation. Even if this does not radically change the main results, it would put the results on a sounder footing, better evaluate sources and sizes of uncertainties, and allow the results given here to be compared more objectively to past and future studies. For this reason, I would not recommend publication in its present form. My recommendations are as follows: 1. Follow the state of the art In the current analysis, evidence for the reconnection X-line is essentially based on looking

for high-speed flows in the vicinity of a high radar spectral width region (e.g., Figure 2a-d) and the X-line extent is estimated from a longitudinal profile of northward velocity at a relatively arbitrary magnetic latitude. In my view this is a rather crude analysis and it should be possible to do this better by estimating the profile of the reconnection electric field itself along the open-closed field line boundary (OCB) and its time evolution following the methodology set out in detail in: Chisham, G., et al. (2008), Remote sensing of the spatial and temporal structure of magnetopause and magnetotail reconnection from the ionosphere, Rev. Geophys., 46, RG1004, doi:10.1029/2007RG000223. Freeman, M. P., G. Chisham, and I. J. Coleman (2007), Remote sensing of reconnection, in Reconnection of Magnetic Fields, edited by J. Birn and E. Priest, chap. 4.6, pp. 217–228, Cambridge Univ. Press, New York. In essence, this method requires the following steps: a. Identify the OCB objectively at as many locations as possible using available datasets and interpolate in space and time where necessary using suitable models, e.g., figures 6, 8, 9, 11 in Chisham et al (2008). b. Estimate the reconnection electric field along the OCB by measuring the electric field component parallel to the boundary (or ExB velocity component perpendicular to it) in the rest frame of the generally moving boundary, e.g., figure 13 in Chisham et al. (2008). c. Plot profiles of the reconnection electric field versus MLT over the time interval of interest. Use the zero crossing locations of these profiles to estimate the MLT extent of reconnection as a function of time, e.g., figure 7 of Pinnock et al., (2003), The location and rate of dayside reconnection during an interval of southward interplanetary magnetic field, Ann. Geophys., 21, 1467–1482. d. Project the MLT extent to the magnetopause using a suitable model to estimate the X-line length and its evolution and to compare with in-situ spacecraft observations of presence or absence of reconnection, e.g., figure 8 of Pinnock et al. (2003). The authors' analysis is only a very crude approximation to this. Particular areas of improvement that I would recommend include:

Response: We thank the reviewer for the detailed comments and instructions. We realize that our view of "X-line" is different from the reviewer's and this seems to have affected the understanding of how an X-line extent should be measured. In our original

terminology we used "magnetic separator" to refer to the global configuration along which reconnection occurs at various rates, and used "X-lines" to refer to regions of strong reconnection, i.e., reconnection bursts, which could activate over a segment of the magnetic separator. The focus of the paper is the latter, as motivated by progresses in recent numerical simulations [Shay et al., 2003; Sheperd and Cassak, 2012]. To avoid confusion, we replace "extent of X-lines" with "extent of reconnection bursts".

The references of X-line extent given by the reviewer provide valuable groundwork of clarifying the scope of this study. We rewrote the first paragraph as "...Reconnection tends to occur at sites of strictly anti-parallel magnetic fields as anti-parallel reconnection [e.g. Crooker, 1979; Luhmann et al., 1984], or occur along a line passing through the subsolar region as component reconnection [e.g. Sonnerup, 1974; Gonzalez and Mozer, 1974]. Evidence shows either or both can occur at the magnetopause and the overall reconnection extent can span from a few up to 40 Re [Paschmann et al., 1986; Gosling et al., 1990; Phan and Paschmann, 1996; Coleman et al., 2001; Phan et al., 2001, 2003; Chisham et al., 2002, 2004, 2008; Petrinec and Fuselier, 2003; Fuselier et al., 2002, 2003, 2005, 2010; Petrinec and Fuselier, 2003; Pinnock et al., 2003; Bobra et al., 2004; Trattner et al., 2004, 2007, 2008, 2017; Trenchi et al., 2008]. However, reconnection does not occur uniformly across this configuration but has spatial variations [Pinnock et al., 2003; Chisham et al., 2008]. The local time extent of reconnection bursts is the focus of this study."

The methodology adopted by our paper has been commonly used for studying reconnection bursts. It is a common approach to measure the flow extent at a latitude poleward of the OCB as the reconnection extent [Goertz et al., 1985; Pinnock et al., 1993, 1995; Provan and Yeoman, 1999; Thorolfsson et al., 2000; McWilliams et al., 2001a, 2001b; Elphic et al., 1990; Denig et al., 1993; Neudegg et al., 1999, 2000; Lockwood et al.. 2001; Wild et al., 2001, 2003, 2007; McWilliams et al., 2004; Zhang et al., 2008]. Based on the snapshots the flow extent did not change much over a 2-3° displacement in latitude. Considering these numerous past works, methodology has

followed a standard approach.

However, we appreciate the reviewer's suggestion and think that it is a good idea to compare our flow velocity profile with the reconnection electric field profile derived following Pinnock et al. [2003], Freeman et al. [2007], Chisham et al. [2008]. We have followed the helpful instructions given by the reviewer and presented our result in Figure S3 based on event #1 (replaced with a new event following the advice of reviewer #1). Details can be found below.

2. Improved estimates of the OCB (step 1a above) a. The authors use a 150 m/s spectral width threshold to estimate the OCB but then apply it rather vaguely by drawing a red contour in figures 2d, 4d, 6d which doesn't match the 150 m/s threshold everywhere. The authors then largely ignore this anyway by using examining the ExB velocity on a fixed latitude circle that is generally poleward of where they say the OCB is. For example, for the first event in section 3.1.2, in lines 293-295 it is said that the OCB is at 77 deg latitude based on the spectral width in figure 2d but in lines 360-366 the 80 deg latitude circle is used as the OCB for the velocity cross-section shown in figure 2f. Similarly, in section 3.2.2, it is 77 deg latitude (lines 390-391) from figure 4d and 79 deg latitude (figure 4 caption) used for figure 4f. And in section 3.2.2, it is 80 deg latitude (figure 6 caption) used for figure 6g,h but the spectral width boundary is unstated and appears to be at lower latitude (at about the projected THA position). b. According to the following references it should be possible to estimate the OCB from spectral widths at a wide range of local times using the method of Chisham and Freeman (2004) and I recommend that this be attempted more carefully and objectively. Chisham, G., and M. P. Freeman (2003), A technique for accurately determining the cusp-region polar cap boundary using SuperDARN HF radar measurements, Ann. Geophys., 21, 983–996. Chisham, G., and M. P. Freeman (2004), An investigation of latitudinal transitions in the SuperDARN Doppler spectral width parameter at different magnetic local times, Ann. Geophys., 22, 1187–1202. Chisham, G., M. P. Freeman, and T. Sotirelis (2004a), A statistical comparison of SuperDARN spectral width

boundaries and DMSP particle precipitation boundaries in the nightside ionosphere, Geophys. Res. Lett., 31, L02804, doi:10.1029/2003GL019074. Chisham, G., M. P. Freeman, T. Sotirelis, R. A. Greenwald, M. Lester, and J.-P. Villain (2005a), A statistical comparison of SuperDARN spectral width boundaries and DMSP particle precipitation boundaries in the morning sector ionosphere, Ann. Geophys., 23,733–743. Chisham, G., M. P. Freeman, T. Sotirelis, and R. A. Greenwald (2005b), The accuracy of using the spectral width boundary measured in off-meridional SuperDARN HF radar beams as a proxy for the open-closed field line boundary, Ann. Geophys., 23, 2599–2604. Chisham, G., M. P. Freeman, M. M. Lam, G. A. Abel, T. Sotirelis, R. A. Greenwald, and M. Lester (2005c), A statistical comparison of SuperDARN spectral width boundaries and DMSP particle precipitation boundaries in the afternoon sector ionosphere, Ann. Geophys., 23, 3645–3654. c. The OCB can also be estimated from other data, such as DMSP particle precipitation. It seems that this data might be available for the events studied, see https://heliophysicsdata.sci.gsfc.nasa.gov/websearch/dispatcher Even if not particularly close in MLT or UT it may be useful as a constraint. d. The T89 model projections of the THA magnetopause crossing to the ionosphere in Figures 4 and 6 appear to agree with the OCB location estimated from the spectral width. It would thus seem reasonable to use the model to estimate the OCB location in the ionosphere at all dayside MLT at this UT. The projected location of THE may be different in these two cases because from Figure 3 there is evidently a rapid outward expansion of the magnetopause from 9.4 RE to 10.2 RE between 1826 and 1828 UT which would need appropriate re-scaling of the model to capture, and in Figure 5 the spacecraft are separated by over 30 min in time and so again the model conditions are probably different. In these cases, and for the figure 2 event, it seems reasonable to explore simple scalings of the T89 model that would fit the magnetopause crossing location of each spacecraft and see if this improves the projected location of the spacecraft with respect to the spectral width boundary. If so, then the model could be used to extrapolate to all dayside MLT. e. Alternatively, a simple offset circle model is commonly a good approximation to the OCB, whose free parameters could be constrained by spectral width and

possibly DMSP data. This would at least be an improvement on assuming a latitudinal circle that is rather unrelated to the spectral width boundary. In all of the above cases, limitations and assumptions can be assessed by error and sensitivity analyses. For example, how are the results 1b-d above affected by changing the inferred boundary by 1 degree say?

Response: Figures S3a-c present the OCB of event #1 around the space-ground con-junction time and longitude. We have identified the OCB more precisely following Chisham and Freeman [2003, 2004] and Chisham et al. [2004, 2005a, 2005b, 2005c] and it is drawn as the dashed black line. The OCB in this event was found nearly along a constant latitude. THEMIS satellite footprints were mapped very closely to the OCB.

3. Take account of the generally moving OCB (step 1b above) As emphasised in the references in 1 above, the reconnection rate is the electric field in the frame of the moving OCB and this can sometimes affect the inference of whether reconnection is occurring or not, e.g., see Figure 13 of Chisham et al. (2008). Some account of this should be taken in the present analysis as it may affect the edges of the inferred reconnection region in particular and hence the FWHM.

Response: Figures S3d-f present time series of the spectral width measurements along beams 4, 7, and 10, as a function of latitude. The time series plot allows us to determine the speed of the OCB motion and we determined the speed at each indi-vidual beam. Note that the OCB motion was longitudinally dependent and was faster around eastern than western beams.

4. Project the ExB velocity perpendicular to the boundary (relevant to step 1c above) Given the strong rotation of the flow seen in figure 2 in particular, consideration should be given of the effect of uncertainties in the assumed orientation of the OCB on the projected flow component across it as this could change the inferred X-line extent.

Response: Figure S3g presents the electric field along the OCB in the frame of the ionosphere (dotted), and in the frame of the OCB (solid). The latter is the reconnection

electric field. The reconnection electric field had essentially the same FWHM as the flow slightly poleward of the OCB (difference being less than the radar spatial resolution). Although the method suggested by the reviewer has its advantage, we note that the process of tracking OCB motion can introduce large uncertainties, especially for our events where the OCB moved very slowly (Figure S1). Given the radar spatial ($\sim 0.3°$) and temporal (2 min) resolution, the speed of OCB has an uncertainty of $\sim 300$ m/s. This results in a signal to noise ratio generally around or even below 1 for the OCB speed, even though we have not yet considered the measurement error associated with spectral widths or the error of using 150 m/s as the OCB threshold in any given event. A similarly poor signal to noise ratio has been found in Chisham et al. [2008]. This would affect the estimate of the electric field and would reduce the confidence of the results. Therefore it is not entirely clear to us whether deriving the reconnection electric field serves as a better methodology for the purpose of our study.

Our study does not discuss the magnitude of the reconnection electric field, but the width is the focus. The flow velocity poleward of the OCB is less affected by the OCB uncertainties. Given that the electric field profiles at the OCB latitude and the flow velocity profile slightly poleward are about the same, that the echoes are more continuous at higher latitudes, and that our approach is consistent with a number of past works cited above, we think that our approach is sufficient to lead to the conclusion.

The above discussion has been clarified in the text as "It is noteworthy mentioning that the velocity profile obtained above approximates to the profile of reconnection electric field along the open-closed field line boundary (details in Figure S3). Reconnection electric field can be estimated by measuring the flow across the open-closed field line boundary in the reference frame of the boundary [Pinnock et al., 2003; Freeman et al., 2007; Chisham et al., 2008]. However, a precise determination of the boundary motion is subject to radar spatial and temporal resolution and for a slow motion like events studied in this paper (Figure S1), the signal to noise ratio is lower than one. For this reason this paper focuses on the velocity profile poleward of the open-closed field

line boundary, which is less affected by the error associated with the boundary. " We did not consider the OCB location beyond the radar FOV because this study is about flows in the satellite-ground conjunction region (not the entire X-line extent). Since the flow FWHM is confined in the radar FOV, our conclusion does not rely on flow or OCB outside the radar FOV.

5. Improved consideration of the temporal evolution The current analyses are strongly biased towards comparisons of magnetopause and ionospheric observations of reconnection at a common instant. Given the uncertainties in how reconnection may evolve at the magnetopause, and the ionospheric response times, it would helpful to repeat the analysis shown in figure 2f, 4f, and 6g,h at some sampling frequency throughout the intervals shown in figures 2e, 4e, and 6e,f. The temporal evolution of los data shown in figures 2e, 4e, and 6e,f are a rather poor proxy by which to estimate the evolution of X-line extent and something similar to figure 7 of Pinnock et al (2003) would be very interesting to see, especially for the inferred complex evolution of the Apr 29 event.

Response: As clarified above, we target reconnection bursts whose extent is by convention measured as the ionospheric flow width. We also focus on the times of satellite magnetopause crossings in order to achieve a space-ground comparison.

6. Discrepancies in magnetopause to ionosphere projection (step 1d above) The magnetopause crossings of spacecraft THA and THD in figure 2, and THE in figure 4 (and possibly figure 6 too) project several degrees of latitude away from the expected OCB location based on spectral width. This suggests that the estimation of X-line extent at the magnetopause from that inferred in the ionosphere will be in error because it is based on the same T89 model that seemingly incorrectly projects the satellite position to the ionosphere. As mentioned in 2d above, it would be helpful to try to estimate the uncertainty by considering whether there is some simple rescaling of the T89 model that would reduce the discrepancy in the magnetopause-to-ionosphere projection. I would also add that the description of the mapping method given in lines 372-376 is too vague to allow others to reproduce your method. It also seems that you use the

same T89 mapping factor of 55 for all three events, which seems questionable, e.g., solar wind dynamic pressure is 50% larger for Apr 19 event. It also implies that the factor is the same for all MLT which is unlikely I think, especially over the 10 Re magnetopause extent inferred for the Apr 29 event. Please could you improve your method description and assess the associated uncertainties.

Response: We would like to clarify that the T89 model is Kp based and does not have solar wind input. Our events all occurred around Kp=2 and that's why the mapping factor is the about same.

In the new Figure #2 (see attachment), the satellite footprints were mapped within the radar FOV and nearly aligned with the OCB. In the Figure 4 event, the 'outward magnetopause motion' does not appear to be due to IMF or solar wind pressure pulses because neither changed substantially. Local distortions of the magnetopause may be a possibility. In any case, there is no known reliable way to modify the model and thus we choose to take the best estimate from the model. In the Figure 6 case, THE crossed the magnetopause later than THA, and at the time of Figure 6 THE was still inside the magnetosphere. THE footprint later on moved to the OCB as the satellite crossed the magnetopause.

As mentioned above, our study does not concern OCB outside the radar FOV. Although we agree that the OCB could be obtained by model magnetopause mapping or addition of DMSP, it does not affect the reconnection burst extent within the radar FOV.

7. I would recommend that you reference and discuss the following first 5 papers in lines 136-141 as these have done a similar comparison of simultaneous reconnection evidence from space and ground to infer X-line length. I would also recommend that you consider the implications of these and the sixth reference to your discussion in section 3.4 as they seem to be relevant to the factors affecting X-line extent (e.g., IMF orientation, component or anti-parallel reconnection, turbulence): Phan, T.D., Freeman, M.P., Kistler, L.M. et al. Earth Planet Sp (2001) 53: 619.

https://doi.org/10.1186/BF03353281 Pinnock, M., G. Chisham, I. J. Coleman, M. P. Freeman, M. Hairston, and J.-P. Villain (2003), The location and rate of dayside reconnection during an interval of southward interplanetary magnetic field, Ann. Geophys., 21, 1467–1482. Coleman, I. J., G. Chisham, M. Pinnock, and M. P. Freeman (2001), An ionospheric convection signature of antiparallel reconnection, J. Geophys. Res., 106, 28,995–29,007. Chisham, G., I. J. Coleman, M. P. Freeman, M. Pinnock, and M. Lester (2002), Ionospheric signatures of split reconnection X-lines during conditions of IMF Bz < 0 and |By|/|Bz|: Evidence for the antiparallel merging hypothesis, J. Geophys. Res., 107(A10), 1323, doi:10.1029/2001JA009124. Chisham, G., M. P. Freeman, I. J. Coleman, M. Pinnock, M. R. Hairston, M. Lester, and G. Sofko (2004b), Measuring the dayside reconnection rate during an interval of due northward interplanetary magnetic field, Ann. Geophys., 22, 4243–4258 Coleman, I. J., and M. P. Freeman (2005), Fractal reconnection structures on the magnetopause, Geophys. Res. Lett., 32, L03115, doi:10.1029/2004GL021779.

Response: We modify the text in Section 3.4 as "...The IMF Bx and By components are known to modify the magnetic shear across the magnetopause and to affect the occurrence location of reconnection. Studies have found that small $|(B_y|)/(|B_z|)$ relates to anti-parallel and large $|(B_y|)/(|B_z|)$ to component reconnection [Coleman et al., 2001; Chisham et al., 2002; Trattner et al., 2007]. Large $|(B_x|)/(|B)|$, i.e. cone angle, also favors formation of high-speed magnetosheath jets [Archer and Horbury, 2013; Plaschke et al., 2013] of a few Re in scale size, resulting in a turbulent magnetosheath environment for reconnection to occur [Coleman, and Freeman, 2005]"

**Fig. 1.** Figure 1a: OMNI IMF condition on Feb 2, 2013. Figure 1b: THE and THA locations projected to the GSM X-Y plane. The inner curve marks the magnetopause and the outer curve marks the bow shock.

[Figure]

**Fig. 2.** Figure 2a: SuperDARN LOS speeds (color tiles) and merged velocity vectors (color arrows) in the Altitude adjusted corrected geomagnetic (AACGM) coordinates. The FOVs of the RKN, INV, and CLY radars ar

**2015 Apr 19**

(a)

IMF (nT)

|B| Bx By Bz

UT  1815  1825  1835

(b)

Y_gsm (Re)

THA
THE

X_gsm (Re)

**Fig. 3.** Figure 3: OMNI IMF condition and THEMIS satellite locations on Apr 19, 2015 in a similar format to Figure 1.

**Fig. 4.** Figure 4. THEMIS and SuperDARN measurements of reconnection bursts on Apr 19, 2015 in a similar format to Figure 2. The velocity time evolution in Figure 4e and the velocity profile in Figure 4f are t

**2015 Apr 29**

(a)

IMF (nT)

|B|  Bx  By  Bz

UT  1820  1840  1900  1920  1940

(b)

Y_gsm (Re)

THA  THE

X_gsm (Re)

**Fig. 5.** Figure 5. OMNI IMF condition and THEMIS satellite locations on Apr 29, 2015 in a similar format to Figure 1.

**Fig. 6.** Figures 6a-d: SuperDARN measurements of reconnection bursts on Apr 29, 2015 in a similar format to Figures 2a-d except that in Figure 6a the color of the CLY color tiles represent LOS speeds towards t

[Figure]

Feb 02, 2013

INV beam0-15

Apr 19, 2015

RKN beam0-15

Apr 29, 2015

INV beam12-15

RKN beam0-1

**Fig. 7.** Figure S1. Location of the open-closed field line boundary (marked by the black dashed line) in the three studied events. The open-closed field line boundary is determined based on the spectral width

Feb 02, 2013

Apr 19, 2015

Apr 29, 2015

**Fig. 8.** Figure S2. Global convection maps of the three studied events. The SHF velocities are shown as color arrows, and the contours of the electric potential are shown as black solid (at the duskside) and d

**Fig. 9.** Figure S3. Reconnection electric along the open-closed field line boundary for the Feb 02, 2013 event. Figures S3a-c: snapshots of spectral width measurements around the space-ground conjunction time

---

## Author Response (AR2)

The author's response has provided clarification, and the changes to the paper have largely helped to convey better the objectives of the study and the reasoning behind the analysis techniques used. Ideally, I would still recommend measuring the reconnection-driven ionospheric flows at the OCB if the data allow, as this is the most direct method of remotely-sensing reconnection (and its extent) from the ionosphere. However, I do accept that (in the absence of good measurements at the OCB) there is still merit to measuring the extent of reconnection bursts from strong ionospheric flows poleward of the OCB, as is the case here. This has been used in many previous studies, even though it represents a simplified approach. However, the authors still need to consider seriously the points made below before the paper is published.

We thank the reviewer for the positive response and have further improved the manuscript according to the reviewer's useful suggestions. The methodology is updated such that the extent of the flow is measured at 1° (as opposed to 2° used previously) poleward of the OCB. We have further added the extent of the reconnection electric field, which is measured right at the OCB, for all events. The two extents (flow & reconnection electric field) show good consistency. We believe that the updated methodology has strengthened the conclusions of the current study.

Major Comments:

(A) Regarding the latitudes chosen to measure the longitudinal extent:

Obviously, the magnetic local time (MLT) extent and width of the flow channel will change as the flow proceeds further into the polar cap. In all three events, the latitude chosen to determine the longitudinal extent (for panels e and f in figures 2 and 4, and e-h in figure 6) is 2 degrees poleward of the estimated location of the OCB. The variation of the magenta lines encompassing the flow in these figures shows that the extent of the reconnection burst flows, and especially their MLT position can change significantly across these 2 degrees.

The authors have strongly defended their decision and provide comments to justify using data at the more poleward latitude, e.g., lines 384-386 – "While this latitude is 2 degrees poleward of the open-closed field line boundary, the shape of the flow did not change much over the 2 degrees displacement and thus still presents the reconnection extent." My response would be, if this statement is true, then why not present the variation at the OCB? Or, at least at a point closer to the OCB. I am not totally convinced, from looking at the data shown, that this statement is necessarily true. If the authors are convinced that the higher latitude (e.g., 80 degrees) is to be used then showing a comparison of the longitudinal extent of the flows (preferably of the poleward component of the SECS flows and not the line-of-sight (LOS) values [see below]) at and 80 degrees (and possibly at other locations in between) is needed to prove the point.

I am still of the opinion that the closer that you can get to actually using the flows at the estimated OCB location, the better.

To incorporate the reviewer's suggestion, we have updated our methodology in the following two aspects. On one hand, we take the measurements 1° (as opposed to 2° used previously) poleward of the OCB to obtain the extent of the flow. The measurements have been smoothed in latitude with a 1° window, following Chisham et al. [2008], to reduce noise and to fill some gaps. On the other hand, we have derived the extent of the reconnection electric field. The derivation is based on the smoothed velocity at the OCB following the method of *Pinnock et al.* [2003], *Freeman et al.* [2007], *Chisham et al.* [2008].

The updated methodology has been applied to all three events. The results show that the two extents agree with each other (see the updated Figures 3g, 5f, and 7f) despite the data gaps in the derived reconnection electric field due to limited echo availability. The description of the methodology and the results have been added at lines 429-479, 533-538, and 643-647 (track-change version).

(B) Regarding the use of LOS flows and the time series of the longitudinal extent:

The time series plots that show the evolution of the extent of the flows are very informative and help to interpret the evolution of each of the events. However, using LOS measurements for this purpose is not a particularly good idea. The beam directions across each radar field-of-view (FOV) vary by ~+/-26 degrees from the central look direction. Hence, comparing LOS measurements from radar beams looking in multiple directions can be seriously misleading.

For example, lines 511-512 – "The velocity at -74 to -30 degrees MLON dropped by 100-200 m/s during 1900-1910 UT, while the velocity at -88 to -74 degrees MLON did not change substantially" – Firstly, in LOS measurements, changes like this can happen just due to slight changes in the direction of the bulk plasma flow relative to the line-of-sight direction. Secondly, these are LOS measurements made by different radars with different look directions. At the join between the measurements from the two radars (between figs 6e and 6f), the look directions of the beams from the two radars differ by more than 90 degrees. Hence, these two radars will measure very different LOS flows at this location. Hence, it is difficult to match together the two figures, and it should not be attempted! It would be much better to show the temporal evolution of the poleward component of the SECS flow here.

In addition, MLON should not be used as one of the axes in these plots. Variations should be plotted against magnetic local time (MLT). This removes changes in the longitude of the flow that are related solely to the rotation of the Earth, which is not relevant to this study.

Hence, my recommendation is that panel e in figs 2, 4, and 6 would be much clearer, less ambiguous, and more easy to interpret if (i) the poleward component of the SECS flow was plotted instead of the LOS velocity, and (ii) MLT was used on the y-axis instead of MLON.

We have updated the figures following the reviewer's advice. The time series plots in Figures 2e, 5e, and 7e now show the northward component of the 2d SECS flow velocity, which gives a more reliable presentation of flow activity without the ambiguities due to radar looking direction. The evolution pattern has not changed much. This is not surprising to see because we had carefully ensured that the previously used LOS data reflect the major flow velocity component. The y axis of those panels has also been changed to MLT.

(C) Regarding the potential effects of IMF Bx and By on the reconnection burst extent:

I don't think that enough events have been observed to allow there to be any significant comment on the effects of Bx and By on the reconnection burst extent. There is not enough evidence to support the conclusions presented in section 3.4. I would consider removing section 3.4.

We agree that more events should be studied to draw a solid conclusion on solar wind condition dependence. We, however, think that it is useful to present the solar wind conditions and to mention the similarities and differences to the extent we can see within the events studied here. We have moved this section to the discussion section to clarify that we are not counting this section as the results of this paper. We have also toned down the conclusion in the last paragraph of the manuscript for consistency.

Minor Comments:

(1) Lines 82-83 – Petrinec and Fuselier (2003) appears twice in this list of references. Deleted one.

(2) Line 116 – "FTEs have been observed to be > or < 2 Re wide in local time" – surely this relates to any size of FTE, it will either be smaller or greater than 2 Re. Hence, I don't get the point of this statement.
Changed to "FTEs have been observed to be on the order of a few Re wide in local time"

(3) Line 198 – Remove 'are' after the [Broll et al. 2017] reference.

Removed.

(4) Lines 327-328 – The phrase "…was confined within the utilized few radar FOVs" would be better written as "…was confined within the FOVs of the radars used".
Corrected (5) Line 553 and Figure 7 – Figure 7 would benefit from the addition of the IMF clock angle variation. The predicted locations of anti-parallel reconnection vary significantly with clock angle, and it is easier to be able to see the clock angle variation without having to visualise the variation based on the variations in IMF By and Bz.
Added. The relative magnitudes of the clock angle is ordered in the same way as the By component.

(6) Lines 569-570 – "Studies have found that small $|By|/|Bz|$ relates to anti-parallel and large $|By|/|Bz|$ to component reconnection" – Significant anti-parallel reconnection still occurs for large $|By|/|Bz|$, but it occurs at higher latitudes on the magnetopause, away from the equatorial plane
We have changed the statement to "Studies have found that at dayside low latitude magnetopause, small $|By|/|Bz|$ relates to anti-parallel and large $|By|/|Bz|$ to component reconnection".
.

Response to reviewer #3

As stated by the title, the goal of this study is to measure the local time extent of magnetopause reconnection bursts using space-ground coordination. This is a very worthwhile scientific goal because knowing the extent of reconnection is important for understanding the geometrical and other factors influencing the reconnection process, which in turn is fundamental to understanding so much of magnetosphere-ionosphere physics and space weather. However, in my opinion, the definition of a reconnection burst and the methodology used to estimate its extent remains too imprecise and inconsistent that the quoted extents of 3, 5, and 11 Re are of questionable scientific value. If this could be improved then I think this would become an excellent and valuable study.

There are really too many detailed points for me to go through so I shall focus on my major concerns:

1. Definition of a reconnection burst. In their response to my first review, the authors say that they are not interested in the extent of non-zero reconnection rate along the magnetic separator but rather the extent of reconnection bursts within it. However, I can find nowhere in the manuscript where a reconnection burst is objectively defined. The implication seems to be that it is a patch of los or poleward component of ionospheric flow above some threshold that is physically distinct from lower los or poleward flow. In practice, a reconnection burst is effectively defined in the paper as a continuous region with los or poleward flow component exceeding half of the peak value. Thus I see no evidence that a reconnection burst is a distinct physical phenomena but merely the highest reconnection rate region of a more extended reconnecting region.

In view of this, I strongly recommend that you do not use the term burst. Instead in the title and elsewhere you should say that you are measuring the local time extent of magnetopause reconnection (not local time extent of magnetopause reconnection bursts) and then clearly state what your definition of local time extent is. At minimum, this could be the definition that you have been using – the region exceeding half the peak reconnection rate value. However, note that for the a Gaussian spatial variation in reconnection rate (e.g., line 387), the FWHM points are at +/-1.18 standard deviations from the peak and thus about 30% of the total reconnection rate lies outside these bounds.

We agree that "burst" may not the best term. This paper focuses on plasma flows produced by reconnection, i.e., reconnection jets, and our previous manuscript implicitly assumed that reconnection jet extent equates the reconnection extent. Reconnection jets correspond to regions of fast generation of open magnetic flux, and as the reviewer suggested, regions of strong reconnection electric field. We admit that weak reconnection may extend over a broader area, but it is the strong reconnection that effectively contributes to the momentum and energy flow within the magnetosphere. This study explores how wide the strong reconnection electric field is. We have now clarified the motivation in the first introduction paragraph as

*"However, reconnection does not occur uniformly across this configuration but has spatial variations [Pinnock et al., 2003; Chisham et al., 2008], and it is the reconnection of high reconnection rates that effectively contributes to the momentum and energy flow within the magnetosphere. Reconnection of high reconnection rates is expected to cause rapid magnetic flux generation and fast reconnection jets. This paper therefore investigates the spatial extent of reconnection through the extents of reconnection jets."*

Terminology changes suggested by the reviewer have been made throughout the text. And the definition of reconnection jets has been clarified in the methodology section.

We understand the reviewer's concern that weak reconnection can extend outside the FWHM, and thus now we also mention the 1-sigma extent for a reference (see the response to the comment below). Here we would like to further clarify our reasoning of using FWHM.

While thresholds for ionosphere flow characterization (half maximum, 1/e or 1 sigma) can seem arbitrary, our choice of half maximum is made for the purpose of a consistency with the definition used for reconnection researches in the magnetosphere. In simulations, Shay et al. [2003] measured the reconnection extent as regions of electron speed above half of the peak electron flow speed during reconnection. In in-situ observations, reconnection jets are defined as regions where the plasma velocity quantitatively agrees (>50%) with the Walen relation. Weaker jets could spread over wider regions along the magnetopause but they are not called reconnection jets. Our case study #1 shows that fast poleward ionospheric flows agree with the Walen relation while slow ionospheric flows do not, which gives the physical distinction between fast and slow ionospheric flows. If a lower threshold (e.g., 1/e or 1 sigma) is used, the width determined by the ionosphere flows may become inconsistent with the magnetosphere observations and/or other past studies. We have added one paragraph addressing the definition of width and its relevant limitations in the methodology section as

*"As seen in our observations presented below, the longitudinal profile of the fast anti-sunward ionospheric flows has a near bell shaped curve. We measure the extent based on full width at half maximum (FWHM) of the profile at 1° poleward of the open-closed field line boundary. This choice of FWHM is analogous to Shay et al. [2003], where the reconnection extent is measured as regions of electron speed above half of the peak electron flow speed during reconnection. The choice is also supported by magnetopause observations, where we find that ionospheric flows with a speed above half of the peak flow speed map to jets consistent with the Walen relation, while those with a speed below map to jets much slower than the Walen relation (Section 3.1). However, it should be noted that the magnitude of the widths is always dependent on the threshold used, and that half maximum is very likely not the only sensible threshold. Using FWHM excludes ionospheric flows with a speed below half of the peak flow speed. Those flows,*

*if related to reconnection, associate with comparatively slow generation of open magnetic flux and low contribution to geomagnetic activity."*

We have also explicitly stated that the measured widths are FWHM in the abstract and conclusion sections.

Another advantage of FWHM is that the Gaussian slope at half maximum is steeper than that for 1/e or 1 sigma, and thus the extent is less subject to measurement errors (1-2 beam width uncertainty as opposed to several beam width for lower thresholds).

Thus, personally, I think it would be helpful to also quote the full width of non-zero reconnection rate. The full width of non-zero reconnection rate is arguably a better measure too because choosing the half-maximum rather than some other fraction (e.g., 1/e) is arbitrary (see point 3 below) whereas non-zero is not, and the relationship of the FWHM to the total reconnection rate contained within it depends on the shape of the reconnection rate spatial variation. That is, I recommend quoting the region over which the reconnection rate exceeds zero within uncertainties (i.e., the difference from zero is statistically significant). If the non-zero region extends beyond the observed region then the quoted value would be a lower bound.

We have derived the distribution of reconnection electric field for all three events (see Figures 3, 5f, and 7f), based on which we estimate the non-zero reconnection extent. For case study #1, we added that

*"As shown in Figure 3g, the profile of the reconnection electric field had a peak in the azimuthal direction with a limited FWHM, and the FWHM is essentially the same as the flow width just poleward of the boundary (difference being less than the radar spatial resolution). This confirms that our measure of the reconnection jet extent is related to the extent of reconnection of high reconnection rates. Regions of high reconnection rates are localized, although those of low reconnection rate (>0 mV/m) can extend over a much broader region. For example, the western boundary of non-zero reconnection rates was located just at the edge of INV FOV (considering the 15 mV/m uncertainty), and the eastern edge extended beyond INV FOV, likely into where the post-noon flow was originated from. A lower estimate of non-zero reconnection rates is therefore ~4 h MLT. It is likely that there were two components of reconnection at different scales: broad and low-rate background reconnection, and embedded high-rate reconnection."*

For case study #2, we added that
*"While the reconnection electric field had data gaps due to the limited coverage and backscatter availability at near range gate, it implies a western boundary of FWHM consistent with the flow slightly poleward of it. This is also the western boundary of non-zero reconnection rates considering the 15-mV/m uncertainty. The eastern boundary extended beyond RKN FOV."*

For case study #3, we added that

*"The reconnection electric field had a similar FWHM to the flow although regions of non-zero reconnection rates again extended beyond the available coverage indicating an overall extent >4 h MLT."*

2. Estimation of the reconnection rate. In my previous review, I recommended that the authors estimate the reconnection rate from the ionospheric electric field in the frame of the generally moving open-closed field line boundary, following the methodology of Chisham et al. (2008). I thank the authors for trying this for the Feb 2013 event. However, the authors relegate this to the supporting information and dismiss this approach in lines 393-401 of the main manuscript and in their response. I really must take issue with the reasoning for this and strongly recommend that the Chisham et al method is used:

Firstly, in the authors' response, they reject the need for the method because they say "our approach is consistent with a number of past works cited above", by which I assume that they mean the 17 references from Goertz et al (1985) through Zhang et al. (2008) that they cite in response to my point 1. However, it should be noted that these works are all 10 or more years old and pre-date the Chisham et al. (2008) method. Thus in my view the state of the art has changed since then and this should be reflected in the standard of data analysis used in this paper.

Secondly, the authors argue that the uncertainty in the estimation of the OCB velocity is large and thus it is reasonable to focus "on the velocity profile poleward of the open-closed field line boundary, which is less affected by the error associated with the boundary". So what the authors are effectively saying is that in some way the los velocity several degrees poleward of the OCB is a better estimate of the magnetopause reconnection rate than estimating the electric field in the moving frame of the OCB. How can this be? No scientifically based arguments are given as to why this should be so. I fear that what the authors are really saying is that they don't want to acknowledge and deal with the inconvenience of observational uncertainties when estimating the local time extent of reconnection (whether a burst or not). In my opinion this is not good science.

If one truly wants to estimate the local time of reconnection then one must be able to identify where the reconnection rate is non-zero. This inevitably requires identifying the OCB, its motion, and the ExB velocity component perpendicular to the OCB. As the authors correctly say, first order spatial differences of the OCB latitude from SuperDARN measurements introduce an uncertainty of 45 km in 2 min, corresponding to an OCB velocity uncertainty of 375 m/s or about 23 mV/m. However this can effectively be reduced somewhat using higher-order differences for the time derivative, or considering a longer sampling interval if this seems appropriate. Either way, this uncertainty has to be taken into account, as detailed in Chisham et al. (2008).

Applying the first-order uncertainty to Figure S3 one would conclude that the -83 to -94 MLON region has a non-zero reconnection rate at the 1 sigma level and this is thus the minimum extent of reconnection. Admittedly the 1 sigma level is not very compelling to a statistician but that is the reality and the scientific method. At least you have quantified the extent for a given confidence level even if that level is low.

The alternative is that one limits oneself to determining the extent of high-speed or non-zero los or poleward flows at given latitude, as you have done, but then one cannot really claim that this is the extent of reconnection in my view.

We thank the reviewer for the discussion. We agree that reconnection electric field is important and that it gives a crucial context for interpreting the flow extent. Therefore the calculation of reconnection electric field has now been conducted for all three events and the results are presented in the main body of the paper. The distribution of the reconnection electric field is very similar to the flow, although there are large data gaps in the reconnection electric field due to limited coverage and backscatter at near range gates around the OCB.

If one is to measure the total extent of reconnection including that of low reconnection rate, the extent is larger than our radar FOV. Our radar FOV size serves as a lower estimate of the overall extent (see our response above). This is acknowledged in the text at lines 472-479, 535-538, and 645-647 (track-change version).

3. Patchy versus extended reconnection. To further emphasise what I believe is the questionable scientific value of the three quoted reconnection extents, I would like to compare the los and SECS velocity profiles shown in figures 2f and 6g. The former is inferred to have a reconnection extent of 13 deg MLON or 3 Re at the magnetopause and the latter 63 deg MLON or 11 Re. Yet I suspect from what I can see in figures 2a and 2b that if the velocity profile shown in figure 2f were extended over the full longitudinal extent of the SuperDARN measurements then it would be similar to that in figure 6g.

Specifically figure 6g has a los velocity maximum at -73 deg MLON. It is clearly above the half-maximum value over a 20 deg MLON region between -87 and -65 deg MLON, at or below the half-maximum value between -55 and -65 MLON and rises again to intermediate values between -65 and -20 deg MLON. (Incidentally -27 MLON shown by the black dotted vertical guideline is not the FWHM point). This is concluded to be an extended reconnection example. Figure 2f has a los velocity maximum at -82 deg MLON and is clearly above the half-maximum value over a 13 deg MLON region -92 and -79 deg MLON before dipping down to just below the half-maximum and likely increasing again above it over an extended region beyond the limit of the plot at -70 deg MLON. This is concluded to be a patchy reconnection example yet I believe the distinction between this and the extended reconnection example depends on a marginal difference in the dip below the half-maximum value in the two cases.

For example, if one had chosen 40% of the maximum rather than 50% then both might have been extended. Or if one chose a slightly lower latitude (closer to the OCB) then I suspect from figure 2b that the dip below the half-maximum in Figure 2f might not be as evident. If there is such a sensitivity in the 'reconnection' extent to the velocity threshold and/or latitude then this casts doubt on the scientific robustness and value of the quoted extents, even without the caveats of point 2 above. Apologies if I am wrong but I'd appreciate seeing los and SECS profiles at different latitudes and over the full MLON range to clear this up. Thanks.

We have expanded the longitudinal/MLT range for this event in Figure 2e. Note that the time series plot now shows the northward component of the 2d SECS velocities, which does not have the ambiguities due to radar looking direction. The measurements are also taken from 1° poleward of the OCB as opposed to 2° used previously. We further present the longitudinal cut of this time series plot around the conjunction time (2135 UT) below. The X axis is the distance from magnetic noon. It can be seen that the two flows were separated by an area of low velocity much below the half maximum. The lowest speed between the two flows was ~10 m/s, only 1-2% of the speed at the peak. This is highly contrasted from the broad extent of the flow in Figure 7f.

We would also like to point out there are other features supporting our differentiation of the two flows. Firstly, as seen in Figures 2a-c, the pre- and post-noon flows became more and more separated and propagate towards more and more different directions as they move away from the cusp. This implies that the two flows are driven by different magnetic tension forces. Hence the velocity dip is not a random velocity fluctuation but really distinguishes reconnection associated with different magnetic field topologies. Secondly, the two flows evolved differently in time. The pre-noon flow persisted for ~30 min while the post-noon flow had a ~10-min lifetime. This implies that the two regions of reconnection have quite different spatial and temporal characteristics, and this is the merit of looking beyond the reconnection electric field distribution.

[Figure]

The above clarification has been added to lines 333-340.

4. Other points. Besides the above major points, I'd also like to mention:
a. It's really difficult to relate the MLON profiles with the FOV maps when you don't put MLON labels on the maps!
We have changed the y axis of Figures 2, 5, 7 to MLT to help readers relate to the 2-D snapshots.

b. I think you might be getting your east and west the wrong way round in some places, such as lines 362-366. You say the eastern boundary is at -82 deg MLON and the western boundary is at -77 deg MLON. But isn't westward in the sense of more negative MLON? As confirmed by THA being westward of THE in Figure 1 and Figure 2e.
Corrected.

c. I felt that the argument involving distinguishing between 200 m/s and 220 m/s spectral widths in lines 334-345 to be doubtful. Firstly, I'm not aware that a simple spectral width threshold corresponding to newly-reconnected field line precipitation has been calibrated (as opposed to the Chisham spectral width boundary method). Secondly, the eastward edge of the pre-noon flow region marked by the magenta line in Figure 2a actually lies through the eastern one of the two dark red spectral width regions in Figure 2d, whereas by your argument shouldn't it lie between them or at the eastern edge of the western dark red region? I think this further supports my argument in point 3 above that this is an extended rather than patchy reconnection region.
The purpose is to point out that there may exist additional structures in the broadly enhanced spectral width area. We have toned down the statement as "*there might exist two dark red regions embedded within the ~200-m/s spectral widths. These two regions*

*had slightly higher spectral widths than the surrounding (by 20-50 m/s) and possibly corresponded to the two flows*".

It actually is not surprising for us to see that the spectral width and the flow did not match exactly. The spectral width is affected by precipitation of electrons and the flow velocity is associated with electric field established by Alfven waves. The two processes are closely related but may not necessarily occur at the exact same instance or location. For the specific event, the region of elevated spectral width seems to be overall displaced to the west of the flow. However, it is also possible that the spatial smoothing of spectral width data (as necessary in inferring the open-closed field line boundary) has contributed to the displacement. This is nevertheless beyond the focus of this study.

d. Why do you not publish all 6 events that you have identified? For example in the supporting information at least as a brief description and summary figure like figure 2f, 4f, 6g for each case. It might help strengthen your conclusions.

We appreciate the reviewer's suggestion but our conclusions are solely based on the three presented events. These events have the clearest, and probably the simplest, flow structures and best space-ground conjunctions. They therefore provide the most convincing evidence among the database. On the other hand, the rest of the events have comparatively small coverage of the flow structures or the reconnection electric field, where an extent cannot be easily obtained. Since those events nevertheless have little relevance to our conclusion, we concern that including them would only distract readers from the main points especially when the paper is already long. We would like to withdraw the statement and only focus on the three presented events.

[revised manuscript text omitted]